# IS COMPLEX QUERY ANSWERING REALLY COMPLEX?

## ABSTRACT

Complex query answering (CQA) on knowledge graphs (KGs) is gaining momentum as a challenging reasoning task. In this paper, we show that the current benchmarks for CQA are not really *complex*, and the way they are built distorts our perception of progress in this field. For example, we find that in these benchmarks most queries (up to 98% for some query types) can be reduced to simpler problems, e.g. link prediction, where only one link needs to be predicted. The performance of state-of-the-art CQA models drops significantly when such models are evaluated on queries that cannot be reduced to easier types. Thus, we propose a set of more challenging benchmarks, composed of queries that *require* models to reason over multiple hops and better reflect the construction of real-world KGs. In a systematic empirical investigation, the new benchmarks show that current methods leave much to be desired from current CQA methods.

## 1 INTRODUCTION

A crucial challenge in AI and ML is learning to perform *complex reasoning*, i.e., solving tasks that involve a number of intermediate steps and sub-goals to be completed. Complex query answering (CQA; Hamilton et al., 2018; Zhang et al., 2021; Arakelyan et al., 2021; Zhu et al., 2022) has emerged as one of the most prominent ways to measure complex reasoning over external knowledge bases, encoded as knowledge graphs (KGs; Hogan et al., 2021). For instance, to answer the query:

"*Which actor performed in a movie filmed in New York City and distributed on Blue Ray?*" ($q_1$)

over a KG such as FreeBase (Bollacker et al., 2008), one would need to *first* intersect the set of movies found on *Blue Ray* and the ones shot in *New York City*, and *then* link these intermediate candidate answers to another entity, i.e., an *actor* participating in it. However, the answers computed in this way may not include entities that are unreachable if missing links are present.

To deal with the unavoidable incompleteness of real-world KGs, ML methods were developed to solve CQA in the presence of missing links. *Neural query answering* models, constituting the current state-of-the-art (SoTA) for CQA (Arakelyan et al., 2021; Zhu et al., 2022; Arakelyan et al., 2023; Ren et al., 2023; Galkin et al., 2024b), map queries and KGs (i.e., entities and relation names) into a unified latent space that supports reasoning. Performance measurements on de-facto-standard benchmarks such as FB15k237 (Toutanova & Chen, 2015) and NELL995 (Xiong et al., 2017) suggest that in recent years such models achieved impressive progress on CQA on queries having different structures, and hence posing apparently different levels of difficulty to be answered.

The difficulty of a benchmark relates to the size and structural complexity[1] of its queries, and several query "types" have been proposed (Ren et al., 2020), each involving a different combination of logical operators—conjunctions, disjunctions, and requiring to traverse a number of missing links that generally increases with the number of logical conditions imposed (Figs. 1 and 2). For instance, the query $q_1$ is an example of a "*2i1p*" query type, since it comprises an intersection of two entity sets *(2i)* followed by a path[2] of length one *(1p)*.

In this paper, we argue that ***the perception of progress on CQA benchmarks has been distorted by implicit assumptions in these benchmarks***. We start by noting how both in FB15k237 and NELL995, the vast majority of queries of a complex type simplifies to one simpler type, thanks to the fact that to

---

[1]Not to be confused with the computational complexity of query answering (Dalvi & Suciu, 2012).

[2]Also referenced as projection in related works (Ren et al., 2023).

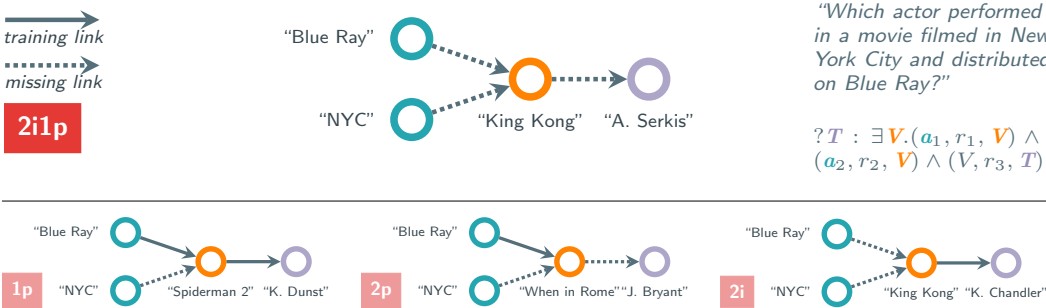

Figure 1: **Query answers are not all equally hard when some links can be found in the training data** as shown for the 2i1p query $q_1$ and fragments of the KG FB15k237, where $r_1 = \text{distributedVia}^{-1}$, $r_2 = \text{locatedIn}^{-1}$ and $r_3 = \text{performedIn}^{-1}$. Its different answers can be obtained by traversing the training graph (continuous line) and predicting the missing links (dotted lines). (Top) Example answer that requires all links to be predicted. (Bottom) Example answers that require only a subset of the links to be predicted and that, therefore, can be reduced to the simpler types 1p, 2p, and 2i (see Sec. 2 and Fig. 2).

answer them one can leverage links already appearing in the training data. Fig. 1 shows an example of how answers to the query $q_1$ on FB15k237– which should require predicting *three* missing links – require the same effort associated to simpler types involving fewer links. In fact, in FB15k237 the answer "K. Dunst" can be retrieved by predicting just one missing link while leveraging the training links. In this sense, a 2i1p query reduces to a "1p" query, i.e., the much simpler task of link prediction (see Sec. 2). Similarly, the answers "J. Bryant" and "K. Chandler" can be retrieved by predicting two links, instead of three. Only the answer "A. Serkis" requires predicting three missing links, and thus satisfies the expected "hardness" associated to the type 2i1p.

We furthermore argue that SoTA performance is inflated likely due to current models memorizing training links. Therefore, we create new CQA benchmarks that comprise only "hard" queries, i.e., queries that cannot be reduced to simpler types by leveraging training links. Then, we perform a thorough re-evaluation of the SoTA models for CQA. Lastly, in order to raise the bar of "complexity" for CQA, we investigate more realistic sampling schemes to generate the train/validation/test splits, and introduce query types that require more reasoning steps in order to be answered.

**Contributions.** After revisiting the CQA task (Sec. 2) and highlighting how query types can simplify at test time (Sec. 3), (**C1**) we show that there is a major data leakage of training links, reporting that up to 99.9% test queries can be reduced to simpler queries, with the majority of them (up to 98%) reduced to "one-step" link prediction problems (Sec. 4). (**C2**) We re-evaluate previous SoTA approaches (Sec. 5), revealing that neural link predictors are relying on memorized information from the training set. Furthermore, we show that the reported hardness of queries involving unions is only apparent. To have a better understanding of why performances are inflated, we introduce CQD-Hybrid (Sec. 5.1), a novel CQA model that combines classical graph-matching with neural link predictors, surpassing the SoTA on existing benchmarks. (**C3**) We create new benchmarks for CQA (Sec. 6), FB15k237+H and NELL995+H from FB15k237 and NELL995, respectively, and ICEWS18+H from the temporal KG ICEWS18 (Boschee et al., 2015), where validation and test links are links that have been added to the KG *after* those in the training, making it more challenging. All these contain only irreducible queries, to which we add the even more challenging query types "4p" (a four-length path) and "4i" (a conjunction of four patterns).

## 2 KGs and Complex Query Answering

**Knowledge graphs.** A KG is a graph-structured knowledge base where knowledge about the world is encoded as relationships between entities. More formally, a KG can be represented as a multi-relational graph $G = (\mathcal{E}, \mathcal{R}, \mathcal{T})$, where $\mathcal{E}$ is a set of entities, $\mathcal{R}$ is a set of relation names, and $\mathcal{T} \subseteq \mathcal{E} \times \mathcal{R} \times \mathcal{E}$ is a set of links or *triples*, where each triple $(s, p, o) \in \mathcal{T}$ represents a relationship of type $p \in \mathcal{R}$ between the subject $s \in \mathcal{E}$ and the object $o \in \mathcal{E}$ of the triple. For instance,

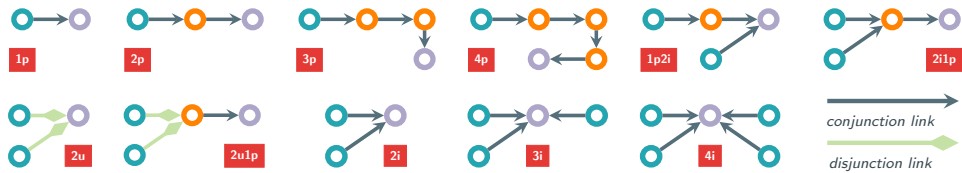

Figure 2: **Query structures we consider**, adapted from Ren & Leskovec (2020) and including *path* (**p**), *intersection* (**i**), *union* (**u**) structures. See Sec. 2 for their logical formulation.

in a KG such as FreeBase (Bollacker et al., 2008), the fact that a movie such as "Spiderman 2" is distributed in Blue Ray can be stored into the triple $(\mathsf{Spiderman2}, \mathsf{distributedVia}, \mathsf{BlueRay})$ or equivalently $(\mathsf{BlueRay}, \mathsf{distributedVia}^{-1}, \mathsf{Spiderman2})$ where $\mathsf{distributedVia}^{-1}$ denotes the inverse relation of $\mathsf{distributedVia}$. Fig. 1 shows examples of fragments from the KG FB15k237.

**Complex Query Answering.** The aim of CQA is to retrieve a set of possible answers to a logical query $q$ that poses conditions over entities and relation types in a KG. Following the literature on CQA (Arakelyan et al., 2021), we consider the problem of answering logical queries with a single *target* variable ($T$), a set of constants including *anchor* entities ($a_1, a_2, \ldots, a_k \in \mathcal{E}$), given relation names ($r_1, r_2, \ldots, r_n \in \mathcal{R}$), and first-order logical operations that include conjunction $\wedge$, disjunction $\vee$ and existential quantification $\exists$. In this work, we do not deal with queries involving negation; however, all of our considerations about the inflated performance of current benchmarks will also transfer to versions of these benchmarks that include them (Ren et al., 2023; Galkin et al., 2020).

Different queries are categorized into different *types* based on the structure of their corresponding logical sentence (Xiong et al., 2017). The idea behind this taxonomy is that queries of the same type share the same "hardness", i.e., the level of difficulty to be answered and different query types correspond to tasks that map to more or less complex reasoning tasks. The simplest CQA task is link prediction (Bordes et al., 2013), i.e., answering a query of the form:

$$?T : (a_1, r_1, T), \tag{1p}$$

that is, given an entity $a_1$ (e.g., $\mathsf{NYC}$) and a relation name $r_1$ (e.g., $\mathsf{locatedIn}^{-1}$), find the entity that when substituted to $T$ correctly matches the link in the KG (e.g., $\mathsf{Spiderman2}$, see Fig. 1). Instead of matching a single link, more complex queries involve matching sub-graphs in a KG (see Fig. 2). Xiong et al. (2017) extend 1p queries, and ask questions that involve traversing sequential paths made of two or three links, i.e.,

$$?T : \exists V_1.(a_1, r_1, V_1) \wedge (V_1, r_2, T), \tag{2p}$$
$$?T : \exists V_1, V_2.(a_1, r_1, V_1) \wedge (V_1, r_2, V_2) \wedge (V_2, r_3, T), \tag{3p}$$

where $V_1, V_2$ denote variables that need to be grounded into entities associated to nodes in the path. Moreover, multiple ground entities can directly participate in a conjunction, e.g., in queries such as:

$$?T : (a_1, r_1, T) \wedge (a_2, r_2, T), \tag{2i}$$
$$?T : (a_1, r_1, T) \wedge (a_2, r_2, T) \wedge (a_3, r_3, T), \tag{3i}$$

which represent the *intersection* of the target entity sets defined over two (2i) or three (3i) links. Path and intersection structures can be combined into more complex queries: for example, the natural language expression for query $q_1$ can be formalized as the formula

$$?T : \exists V_1.(a_1, r_1, V_1) \wedge (a_2, r_2, V_1) \wedge (V_1, r_3, T), \tag{2i1p}$$

involving one intersection followed by a one-length path.[3] See also the example in Fig. 1. By inverting the order of operations, we obtain the query type "1p2i":

$$?T : \exists V_1.(a_1, r_1, V_1) \wedge (V_1, r_2, T) \wedge (a_2, r_3, T). \tag{1p2i}$$

Similarly to the introduction of conjunctions, we can consider disjunctions in queries, realizing the *union* query types which can be answered by matching one link or the other, such as

$$?T : (a_1, r_1, T) \vee (a_2, r_2, T), \tag{2u}$$

---

[3]In previous works (Ren et al., 2020) this query type was referred to as "ip". We explicitly mention the number of steps involved in a path or conjunction, as this is a factor of complexity. Analogously, "pi", "u" and "up" queries from Ren et al. (2020) are now 1p2i, 2u and 2u1p.

or by combining the union with the previous query types, e.g., obtained by combining 2u and 1p:

$$?T : \exists V_1.((a_1, r_1, V_1) \vee (a_2, r_2, V_1)) \wedge (V_1, r_2, T). \tag{2u1p}$$

Note that despite their dissimilar syntaxes and the different sub-graphs defining possible solutions (Fig. 2), the query type 2u should be as difficult as 1p, as to answer the first it suffices to match a single link correctly. Similarly, 2u1p is as complex as 2p. The fact that 2u and 2u1p are reported to be harder to solve in practice than 1p and 2p (Ren et al., 2020) is due to the way standard benchmarks are built, which we discuss in Sec. 4, and the way in which CQA is evaluated, discussed next.

**Standard evaluation.** Given a KG $G$ and a logical query $q$ from one of the types described above, answering $q$ boils down to a graph matching problem (Hogan, 2020) if we assume that all the meaningful links are already in $G$. Instead, if $G$ is incomplete, we will need to predict missing links while answering $q$. Many ML approaches to CQA, reviewed in Sec. 5, therefore assume a distribution over *possible* links (Loconte et al., 2023), requiring *probabilistic reasoning*. To evaluate them, standard benchmarks such as FB15k237 and NELL995 artificially divide $G$ into $G_{\text{train}}$ and $G_{\text{test}}$, treating the triples in the latter as missing links. This splitting process is done uniformly at random, a procedural choice that can alter the measured performance, as we discuss next.

CQA is generally treated as a ranking task, counting how many true candidate answers to a query are ranked higher than non-answer ones. Denoting the rank of a ***query answer pair*** $(q, t)$ by $\text{rank}(q, t)$, the performance for each query type is calculated as the mean reciprocal rank (MRR), i.e.,

$$|\mathcal{Q}|^{-1} \sum_{q \in \mathcal{Q}, t \in \mathcal{E}_q} |\mathcal{E}_q|^{-1} \text{rank}(q, t), \tag{1}$$

where $\mathcal{Q}$ denotes the set of test queries of the considered type, and $\mathcal{E}_q$ is the set of candidate answer entities for each query $q \in \mathcal{Q}$. This average, across queries and answers, assumes that every query answer pair having the same query type is equivalently hard, which is not the case. In fact, we show that certain query answer pairs can be easier to retrieve if links from the training data leak into the model (Fig. 1), and that the distribution of the query answer pairs in the existing benchmarks is very skewed towards those involving a single missing link (Table 1). Thus, computing Eq. (1) without understanding what is the benchmark distribution, distorts the perception of performance gains.

## 3 What is the real "Hardness" of CQA in incomplete KGs?

As discussed in the previous section, the perceived complexity of a query is related to the graph structure associated to its query type (Fig. 2): queries containing more hops/existentially-quantified variables are more challenging, e.g., a 3p query is harder than a 2p query. In this section, we give an alternative perspective on the difficulty of answering queries that take into account the information coming from the training data, and that might have leaked into a learned model. We argue that predicting links that are truly missing, i.e., not accessible to a learned model, is actually what makes a query "hard". To do so, we formalize the notion of a reasoning tree for a query answer pair, and then define how we determine the practical hardness of a query answer pair.

Given a query $q$ and every answer set of candidate answers $t \in \mathcal{E}_q$ in its candidate set, we define the ***reasoning tree*** of each query answer pair $(q, t)$, as the directed acyclic graph starting from the anchor entities of $q$ to the target entity $t$, whose relational structure matches the query graph. Fig. 1 provides examples of different reasoning trees for four different answers to the same query. There, we highlight whether a link belongs to $G_{\text{train}}$ or not, i.e., it is a missing link. We assess the hardness of each answer $t \in \mathcal{E}_q$, by analyzing the composition of the reasoning tree required to predict $t$. As to answering a $(q, t)$ pair, multiple reasoning trees are possible, so we select the one with the smallest number of missing links and the fewest number of hops.

A $(q, t)$ pair is ***trivial*** if the answer $t$ can be entirely retrieved from $G_{\text{train}}$, i.e., there exists at least one reasoning tree where all the links in it are seen during training. This type of answer does not need probabilistic inference and hence is filtered out from current CQA benchmarks (Ren & Leskovec, 2020), which only consider non-trivial $(q, t)$ pairs, which we call ***inference-based*** pairs. However, inference-based pairs do not need all links in their reasoning tree to be predicted as, by definition, it is sufficient that at least one link in the tree is missing. Therefore, we define a $(q, t)$ pair to be a ***partial-inference*** pair if its reasoning tree contains at least one link in $G_{\text{train}}$ and at least one link present in $G_{\text{test}}$. Alternatively, a query answer pair whose reasoning tree contains only links present

Table 1: **The great majority of query answer pairs are of the much easier partial-inference type (non-diagonal), rather than of the harder full-inference type (diagonal).** For each query type (as rows) we show the percentage of the query answer pairs that can be reduced to an easier query type (as columns) for datasets FB15k237 and NELL995. Most of the complex queries can be reduced to simple link prediction queries (i.e., 1p). We denote as '-' those reductions that are not possible given the sub-graph structure induced by the query type.

| | FB15k237 | | | | | | | | | NELL995 | | | | | | | | |
|---|---|---|---|---|---|---|---|---|---|---|---|---|---|---|---|---|---|---|
| | 1p | 2p | 3p | 2i | 3i | 1p2i | 2i1p | 2u | 2u1p | 1p | 2p | 3p | 2i | 3i | 1p2i | 2i1p | 2u | 2u1p |
| 1p | 100 | - | - | - | - | - | - | - | - | 100 | - | - | - | - | - | - | - | - |
| 2p | 98.1 | 1.9 | - | - | - | - | - | - | - | 97.6 | 2.4 | - | - | - | - | - | - | - |
| 3p | 97.2 | 2.7 | 0.1 | - | - | - | - | - | - | 95.6 | 4.3 | 0.1 | - | - | - | - | - | - |
| 2i | 96.0 | - | - | 4.0 | - | - | - | - | - | 94.0 | - | - | 6.0 | - | - | - | - | - |
| 3i | 91.6 | - | - | 8.2 | 0.2 | - | - | - | - | 87.4 | - | - | 12.1 | 0.5 | - | - | - | - |
| 1p2i | 86.8 | 1.0 | - | 12.0 | - | 0.2 | - | - | - | 49.5 | 0.6 | - | 49.0 | - | 0.9 | - | - | - |
| 2i1p | 96.7 | 1.8 | - | 1.4 | - | - | 0.1 | - | - | 96.2 | 2.4 | - | 1.2 | - | - | 0.2 | - | - |
| 2u | 0.0 | - | - | - | - | - | - | 100 | - | 0.0 | - | - | - | - | - | - | 100 | - |
| 2u1p | 98.3 | 0.0 | - | - | - | - | - | 1.6 | 0.1 | 98.5 | 0.0 | - | - | - | - | - | 1.4 | 0.1 |

in $G_{\text{test}}$ is called a ***full-inference*** pair. We claim that predicting an answer of a full-inference kind is harder than doing that for a partial-inference one, as discussed next.

To predict a partial-inference pair $(q, t)$, a ML model that has ***explicit*** access to $G_{\text{train}}$ has to solve a simpler task to answer $q$, as only a subset of the links in the reasoning tree are missing and need to be predicted. As such, the $(q, t)$ pair can be simplified to $(q', t)$ pair, where the query $q'$ is of an easier query type than $q$. Fig. 1 shows one example of how a query answer pairs of type 2i1p from FB15k237 are reduced in practice to the simpler types 1p, 2i and 2p. We will build our hybrid solver in Sec. 5.1 leveraging this intuition. Note that this advantage applies also to ML models that have ***implicit*** access to $G_{\text{train}}$, e.g., by having memorized the triples during training, a common phenomenon for many neural link predictors (Nickel et al., 2014). We confirm this in Sec. 5, showing that the performance of all SoTA models for partial-inference pairs is comparable to the performance of full-inference pairs *but of simpler types*. We next analyze how many partial vs full-inference queries the current benchmarks contain.

## 4    HOW MANY "COMPLEX" QUERIES IN CURRENT CQA BENCHMARKS?

In this section, we systematically analyze the practical hardness of queries from very popular CQA benchmarks and answer the following research question: **(RQ1)** *What is the proportion of query answer pairs that can be classified as full-inference rather than the easier partial-inference?*

For this purpose, we consider the CQA benchmarks generated from the KG FB15k237 (based on Freebase) and NELL995 (based on NELL systems (Carlson et al., 2010)) as they are the most used to evaluate SoTA methods for CQA (Ren et al., 2020; Ren & Leskovec, 2020; Arakelyan et al., 2021; Zhang et al., 2021; Zhu et al., 2022; Arakelyan et al., 2023).[4]

**Complex queries can be reduced to much simpler types.** We group the testing query answer pairs $(q, t)$ into query types (Sec. 2), and we further split them based on whether they can be reduced to simpler types after observing the training links in $G_{\text{train}}$ (Sec. 3). Table 1 shows that for both FB15k237 and NELL995 the vast majority of $(q, t)$ pairs can be reduced to simpler types. For FB15k237, 86.8% to 98.3% of $(q, t)$ pairs can be reduced to 1p queries, while only 0.1% to 4% require full inference. Similarly, for NELL995, 49.5% to 98.5% of $(q, t)$ pairs map to 1p queries, and only 0.1% to 6% to full inference. For instance, 96.7% of 2i1p $(q, t)$ pairs in FB15k237 can be reduced to 1p queries, 1.8% to 2p, and 1.4% to 2i. However, only 0.1% of these pairs cannot be reduced to any other $(q', t)$ pair, i.e., they require full inference in order to be predicted. The only exceptions to this trend are $(q, t)$ pairs where $q$ has a 1p or 2u structure which, by definition, only require the prediction of a single link and therefore cannot be reduced by any other query type.

---

[4]We do not consider FB15k (Bordes et al., 2013) as it suffers from data leakage (Toutanova & Chen, 2015).

Table 2: **SoTA performance drops significantly on full-inference query answer pairs.** For each query type (rows), we show the MRR achieved by SoTA methods on full-inference query answer pairs only (diagonal), and compare it with the MRR achieved on all available testing queries (column "all") and on the queries that can be reduced to simpler types (other columns). We highlight in bold how the best ranking model changes when we consider the full-inference query answer pairs only. See Tables A.3 and A.4 for results including hybrid solvers.

| Query type | method | FB15k237 | | | | | | | | NELL995 | | | | | | | |
|---|---|---|---|---|---|---|---|---|---|---|---|---|---|---|---|---|---|
| | | all | 1p | 2p | 3p | 2i | 3i | 1p2i | 2i1p | all | 1p | 2p | 3p | 2i | 3i | 1p2i | 2i1p |
| 1p | GNN-QE | 42.8 | 42.8 | - | - | - | - | - | - | 53.6 | 53.6 | - | - | - | - | - | - |
| | ULTRAQ | 40.6 | 40.6 | - | - | - | - | - | - | 38.9 | 38.9 | - | - | - | - | - | - |
| | CQD | **46.7** | **46.7** | - | - | - | - | - | - | **60.4** | **60.4** | - | - | - | - | - | - |
| | ConE | 41.8 | 41.8 | - | - | - | - | - | - | 60.0 | 60.0 | - | - | - | - | - | - |
| 2p | GNN-QE | **14.7** | 14.8 | 4.7 | - | - | - | - | - | 17.9 | 18.2 | 6.1 | - | - | - | - | - |
| | ULTRAQ | 11.5 | 11.5 | 4.2 | - | - | - | - | - | 11.2 | 11.5 | 4.6 | - | - | - | - | - |
| | CQD | 13.2 | 13.3 | 3.5 | - | - | - | - | - | **22.0** | 22.3 | **7.6** | - | - | - | - | - |
| | ConE | 12.8 | 12.8 | **5.2** | - | - | - | - | - | 16.0 | 16.3 | 7.2 | - | - | - | - | - |
| 3p | GNN-QE | **11.8** | 12.0 | 4.4 | **4.8** | - | - | - | - | **15.2** | 15.1 | 8.1 | 3.4 | - | - | - | - |
| | ULTRAQ | 8.9 | 9.0 | 4.6 | 4.4 | - | - | - | - | 9.7 | 9.8 | 5.1 | 4.1 | - | - | - | - |
| | CQD | 7.8 | 7.8 | 3.4 | 1.8 | - | - | - | - | 13.4 | 12.8 | 7.8 | **8.5** | - | - | - | - |
| | ConE | 11.0 | 11.0 | 5.4 | 2.6 | - | - | - | - | 13.8 | 13.8 | 8.2 | 6.5 | - | - | - | - |
| 2i | GNN-QE | **38.3** | 39.3 | - | - | 4.0 | - | - | - | 40.0 | 41.4 | - | - | 3.6 | - | - | - |
| | ULTRAQ | 35.7 | 36.7 | - | - | 3.4 | - | - | - | 36.3 | 37.5 | - | - | 3.2 | - | - | - |
| | CQD | 35.0 | 35.8 | - | - | **7.3** | - | - | - | **42.2** | 43.3 | - | - | 6.9 | - | - | - |
| | ConE | 32.6 | 33.3 | - | - | 5.5 | - | - | - | 39.6 | 40.6 | - | - | **8.0** | - | - | - |
| 3i | GNN-QE | **54.1** | 56.0 | - | - | 10.9 | 5.2 | - | - | 50.9 | 53.6 | - | - | 10.9 | 2.2 | - | - |
| | ULTRAQ | 51.0 | 52.9 | - | - | 9.7 | 4.3 | - | - | 47.7 | 50.0 | - | - | 11.3 | 1.5 | - | - |
| | CQD | 48.5 | 49.6 | - | - | 17.0 | **6.0** | - | - | **51.8** | 53.9 | - | - | 14.4 | 3.6 | - | - |
| | ConE | 47.3 | 48.3 | - | - | 15.6 | 4.0 | - | - | 50.2 | 51.9 | - | - | 18.3 | **4.8** | - | - |
| 1p2i | GNN-QE | **31.1** | 32.8 | 15.5 | - | 5.7 | - | 9.1 | - | 29.1 | 42.7 | 21.4 | - | 12.0 | - | **10.2** | - |
| | ULTRAQ | 29.6 | 31.5 | 19.2 | - | 4.2 | - | 7.4 | - | 25.1 | 39.0 | 25.6 | - | 8.2 | - | 7.0 | - |
| | CQD | 27.5 | 28.7 | 13.4 | - | 7.1 | - | 9.0 | - | **31.5** | 44.0 | 25.7 | - | 14.5 | - | **12.6** | - |
| | ConE | 25.5 | 26.6 | 13.9 | - | 5.4 | - | **9.7** | - | 26.1 | 38.5 | 25.2 | - | 10.5 | - | 8.6 | - |
| 2i1p | GNN-QE | 18.9 | 19.2 | 8.2 | - | 6.2 | - | - | 3.4 | 20.5 | 20.8 | 16.3 | - | 14.6 | - | - | 20.7 |
| | ULTRAQ | 18.6 | 18.9 | 8.5 | - | 5.1 | - | - | **8.1** | 15.6 | 16.5 | 9.5 | - | 9.6 | - | - | 11.4 |
| | CQD | **20.7** | 21.0 | 10.5 | - | 6.7 | - | - | 7.6 | **25.8** | 25.9 | 21.2 | - | 26.3 | - | - | **23.6** |
| | ConE | 14.0 | 13.8 | 9.7 | - | 12.8 | - | - | 5.6 | 17.6 | 17.7 | 16.4 | - | 25.7 | - | - | 19.5 |

**Non-existing links for union queries.** Additionally, we discovered that there are *non-existing* links, i.e., links that are not in the original KG $G$ and hence neither in $G_{\text{train}}$ or $G_{\text{test}}$, in both FB15k237 and NELL995 in the reasoning trees of queries involving unions, i.e., the 2u and 2u1p types. Fig. A.1 shows some examples. These links violate our definitions of inference $(q, t)$ pairs, hence, we filter them out, and report only *filtered* $(q, t)$ pairs in Table 1. More crucially, these non-existing links can alter the performance of solvers for 2u and 2u1p types, as we discuss in the next section.

> **Takeaway 1.**
>
> The de-facto-standard benchmarks for CQA do not provide a clear picture of the ability of a model to solve complex queries, but rather simple link prediction tasks. We discourage reporting average performance per query type and suggest a stratified analysis as in Table 1.

## 5 How do SoTA Models Perform on Full-Inference Queries?

In this section, we re-evaluate a number of SoTA methods for CQA and answer to the following question: **(RQ2)** *How do these approaches perform on partial-inference and full-inference query answer pairs?* Furthermore, we leverage our query hardness distinction (Sec. 3) to devise a model class that explicitly leverages the links stored in the training KG to solve CQA tasks.

**SoTA CQA methods.** Over the years, a significant number of neural models have been proposed for solving the CQA task. Guu et al. (2015) propose compositional training for embedding methods

Table 3: **Queries of type 2u are less hard than previously thought** in terms of MRR by SoTA methods, with our filtered 2u queries having comparable performance to 1p, as discussed in Sec. 2, in contrast with the results of the 2u queries originally used (2u all).

| method | FB15k237 | | | NELL995 | | |
|---|---|---|---|---|---|---|
| | 1p | 2u all | 2u filtered | 1p | 2u all | 2u filtered |
| GNN-QE | 42.8 | 16.2 | **40.7** | 53.6 | 15.4 | 34.8 |
| ULTRAQ | 40.6 | 13.2 | 33.6 | 38.9 | 10.2 | 21.3 |
| CQD | **46.7** | **17.6** | 32.7 | **60.4** | **19.9** | **35.9** |
| ConE | 41.8 | 14.9 | 29.9 | 60.0 | 14.9 | 28.2 |

to predict answers for path queries. GQE (Hamilton et al., 2018) learns a geometric intersection operator to answer conjunctive queries in embedding space; this approach was later extended by Query2Box (Ren et al., 2020), BetaE (Ren & Leskovec, 2020), and GNN-QE (Zhu et al., 2022). FuzzQE (Chen et al., 2022) improves embedding methods with t-norm fuzzy logic, which satisfies the axiomatic system of classical logic. Some recent works such as HypE (Choudhary et al., 2021) and ConE (Zhang et al., 2021) use geometric interpretations of entity and relation embeddings to achieve desired properties for the logical operators. Other solutions to CQA combine neural methods with symbolic algorithms. For example, EmQL (Sun et al., 2020) ensembles an embedding model and a count-min sketch, and is able to find logically entailed answers, while CQD (Arakelyan et al., 2021; 2023) extends a pretrained knowledge graph embedding model to infer answers for complex queries. In this work, we consider four representative approaches that significantly differ in their methodological designs and yield SoTA results compared to other models in their class. We consider the following models: (1) Cone Embeddings (ConE; Zhang et al., 2021) is a geometry-based complex query answering model where logical conjunctions and disjunctions are represented as intersections and union of cones. (2) Graph Neural Network Query Executor (GNN-QE; Zhu et al., 2022) decomposes a complex first-order logic query into projections over fuzzy sets. (3) Continuous Query Decomposition (CQD; Arakelyan et al., 2021; 2023), reduces the CQA task to the problem of finding the most likely variable assignment, where the likelihood of each link (1p query) is assessed by a neural link predictor, and logical connectives are relaxed via fuzzy logic operators. (4) Ultra-Query (ULTRAQ; Galkin et al., 2024b) is a foundation model for CQA inspired by Galkin et al. (2024a) where links and logical operations are represented by vocabulary-independent functions which can generalize to new entities and relation types in any KG.

**SoTA methods perform significantly worse on full-inference queries.** We re-evaluate the SoTA methods mentioned above and stratify their performances based on the type of partial- and full-inference queries. Where available, we reused the pre-trained models for their evaluation. Otherwise, we re-produced the SoTA results using the hyperparameters provided by the authors, listed in App. D.1. Table 2 presents the results in terms of MRR for the testing query answer pairs in the existing benchmarks FB15k237 and NELL995, along with the different $(q', t)$ pairs they can be simplified to when observing the training KG, grouped by each query type. There, the performance of all SoTA methods consistently drops for query answer pairs that have a higher number of missing links in their reasoning tree. Furthermore, Table 2 compares the MRR computed on all the available queries originally used in the benchmarks (column "all"), and the scores on those query types that they can be reduced to. *There is a high similarity in MRR between the columns "all" and "1p" that is evidence that the good performances of CQA methods are explained by their link prediction performances*, as a very high percentage of queries in fact are reduced to 1p queries (see Table 1).

However, in a few cases it happens that full-inference results are higher than some partial-inference; in such cases, the reason is to be found in the fact that full-inference $(q, t)$ pairs are very scarce (see Table 1). We rule out that this is due to the influence of the number of existing entities bounding to existential quantified variables. For example we report such an analysis for CQD in Figs. A.3 and C.2. Moreover, we found that no anchor entities nor relation names is predominant in both benchmarks, as shown in A.5.

**Non-existing links for union queries give a false sense of hardness.** We remark that for union queries, we filter query answer pairs, as discussed in Sec. 4, removing those with non-existing links in their reasoning trees. In Table 3, we show that if we do not do that, and consider also the pairs with non-existing links (denoted as "2u all"), MRR performance greatly drops. In this way, we reproduce the low performance for 2u queries that the original SoTA baselines reported in their papers. However, as Table 3 shows, this is just an artifact due to including non-filtered pairs while

Table 4: **Exploiting the links in the training KG during inference boosts MRR** on existing benchmarks, as shown for our CQD-Hybrid model compared to previous SoTA, for several query types. CQD-Hybrid always achieves higher MRR scores when compared to CQD, and outperforms more sophisticated baselines 10/14 times. Best values in bold and second best underlined. For the stratified comparison of CQD-Hybrid with previous SoTA, please refer to the Tables A.3 and A.4.

| | FB15k237 | | | | | | | NELL995 | | | | | | |
| Method | 2p | 3p | 2i | 3i | 1p2i | 2i1p | 2u1p | 2p | 3p | 2i | 3i | 1p2i | 2i1p | 2u1p |
|---|---|---|---|---|---|---|---|---|---|---|---|---|---|---|
| GNN-QE | 14.7 | **11.8** | **38.3** | **54.1** | 31.1 | 18.9 | 9.7 | 17.9 | 15.2 | 40.0 | 50.9 | 29.1 | 20.5 | 8.8 |
| ULTRAQ | 11.5 | 8.9 | 35.7 | 51.0 | 29.6 | 18.6 | 7.3 | 11.2 | 9.7 | 36.3 | 47.7 | 25.1 | 15.6 | 8.4 |
| CQD | 13.2 | 7.8 | 35.0 | 48.5 | 27.5 | 20.7 | **10.5** | 22.0 | 13.4 | 42.2 | 51.8 | 31.5 | 25.8 | 16.0 |
| ConE | 12.8 | 11.0 | 32.6 | 47.3 | 25.5 | 14.0 | 7.4 | 16.0 | 13.8 | 39.6 | 50.2 | 26.1 | 17.6 | 11.1 |
| **CQD-Hybrid (ours)** | **15.0** | 11.0 | 37.6 | 52.7 | **31.2** | **24.0** | 10.3 | **23.8** | **17.8** | **44.2** | **57.8** | **33.2** | **28.4** | **16.6** |

computing Eq. (1). With our filtered pairs, instead, we recover the performance of 1p queries, as expected (Sec. 2). Similar conclusions can be drawn for other union queries as reported in Table A.2.

## 5.1 BUILDING AN HYBRID REASONER TO EXPLOIT TRAINING LINKS

So far, we have assessed that current benchmarks are skewed towards easier query types (Table 1) and thus the perceived progress of current SoTA CQA methods boils down to their performance on 1p queries (Table 2). To have an undistorted view of this progress, in the next section, we will devise a benchmark that allows to precisely measure model performance on full-inference queries *only*. Nevertheless, we argue that in a real-world scenario one has to perform reasoning over *both* existing links *and* in the presence of missing ones. Therefore, it is worth to evaluate CQA over partial-inference queries, however in a stratified analysis and accounting for the disproportion of 1p links (see Takeaway 1). Next, we discuss how to build an hybrid solver that exploits both link types.

Current SoTA CQA methods might implicitly exploit this aspect at test time if they are able to memorize well the entire training KG. However, there is no guarantee that this is the case, especially for less parameterized models. If one is interested in hybrid inference on real-world KGs, discarding the information in $G_{\text{train}}$ is wasteful. To this end, *hybrid solvers* such as QTO (Bai et al., 2023) and FIT (Yin et al., 2024) have been developed. They explicitly retrieve existing links from $G_{\text{train}}$ when answering a query at test time. Inspired by them, and to evaluate our hypothesis, we create a light-weight hybrid solver, named ***CQD-Hybrid***, that is a variant of CQD that uses a pre-trained link predictor to compute scores for the answer entities of 1p queries only, denoting the unnormalized probability of that single link to exist (Loconte et al., 2023). Then, assignments to existentially quantified variables in a complex query are greedily obtained by maximizing the combined score of the links, computed by replacing logical operators in the query with fuzzy ones (van Krieken et al., 2022). A complete assignments to the variables (and hence to the target variable $T$), gives us an answer to the query. In our CQD-Hybrid, we assign the maximum score to those links that are observed in $G_{\text{train}}$. That is, training links will have a higher score than the one that the link predictor would output, hence effectively steering the mentioned greedy procedure at test time. We report hyperparameters and implementation details of CQD-Hybrid in App. D.1. This is the only minimal change we apply to CQD to test if its performance can depend on memorizing training triples. Note that this is different from QTO and FIT, which involve more sophisticated steps such as score calibration and a forward/backward update stage, that can boost performance. In our experiments, we found CQD-Hybrid was consistently boosting performance in terms of MRR w.r.t. CQD and other non-hybrid baselines, as reported in Table 4, which shows aggregated performance for space reasons. A stratified comparison is reported in the appendix in Tables A.3 and A.4.

> **Takeaway 2.**
>
> FB15k237 and NELL995 are not suitable to precisely assess the capability of CQA methods to answer full-inference queries, resulting in highly inflated performance, as shown in Table 2. However, they can be used to evaluate the performance of *hybrid* models , such as QTO (Bai et al., 2023), FIT (Yin et al., 2024), and CQD-Hybrid, that explicitly exploit existing triples to predict the query answers.

Figure 3: **MRR of SoTA on the new benchmarks (in orange) is significantly lower than the old ones (in blue).** As expected from the hardness analysis presented in Sec. 2, the performance on the new benchmark for 2u are similar to 1p, and the one of 2u1p are similar to 2p.

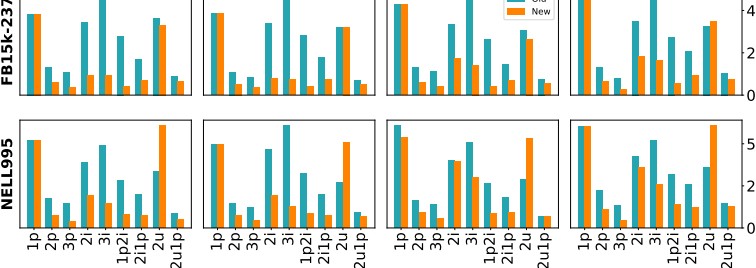

## 6 NEW BENCHMARKS FOR "TRULY COMPLEX" QUERY ANSWERING

In this section, we answer to the following question: (**RQ3**) *How can we construct a set of CQA benchmarks that let us measure truly challenging queries?* To do so, we present a new set of CQA benchmarks based on the previously used FB15k237 and NELL995, as well as a novel one we design out of temporal knowledge graph ICEWS18 to question the current way to split the original KG $G$ into $G_{\text{train}}$ and $G_{\text{test}}$. Then, we evaluate current SoTA methods on these new benchmarks as to establish a strong baseline for future works.

**Building new CQA benchmarks.** We generate our benchmarks to comprise only full-inference question answer pairs, filtering out all partial-inference (and trivial) ones from the evaluation. To this end, we modify the algorithm of Ren et al. (2020) to ensure that no training links are present in the reasoning trees of any $(q, t)$ pair we generate. To raise the bar of "complexity" in CQA, we introduce two query types that, in their full-inference versions, are harder than simpler types by design. Specifically, we design "4p" queries, for a sequential path made of four links, i.e.,

$$?T : \exists V_1, V_2, V_3.(a_1, r_1, V_1) \wedge (V_1, r_2, V_2) \wedge (V_2, r_3, V_3) \wedge (V_3, r_4, T), \tag{4p}$$

and "4i", which represents the intersection of the target entity sets defined over four links, i.e.

$$?T : (a_1, r_1, T) \wedge (a_2, r_2, T) \wedge (a_3, r_3, T) \wedge (a_4, r_4, T). \tag{4i}$$

We name the *harder* versions of FB15k237 and NELL995, FB15k237+H and NELL995+H, each comprising 50.000 full-inference test and validation $(q, t)$ pairs for all query types apart from 1p queries, for which we keep the same set of the old benchmark (this being already full-inference), and for the new types 4p and 4i, for which we have 10.000 of them. More details in App. C.

**Non-uniform at random splits.** For FB15k237+H and NELL995+H, we keep the existing data splits of $G_{\text{train}}$ and $G_{\text{test}}$. However, to evaluate the impact of this artificial splitting process, we adopt a more realistic one for ICEWS18+H, where $G_{\text{train}}$ contains present links and we might want to predict the future links contained in $G_{\text{test}}$. To this end, we leverage the temporal information in ICEWS18 by (1) ordering the links based on their timestamp; (2) removing the temporal information, thus obtaining normal triples; and (3) selecting the train set to be the first temporally-ordered 80% of triples, the valid the next 10%, and the remaining to be the test split. If the same fact appears with multiple temporal information, we retain only the link with the earliest timestamp. For detailed statistics about the splits please refer to App. E.

**How do SoTA methods fare on our new benchmarks?** For FB15k237+H and NELL995+H we re-used the pre-trained models used for the experiments in Sec. 4, while for the newly created ICEWS18+H we trained GNN-QE, ConE, and the link predictor used in CQD and QTO from scratch. As this is not necessary for ULTRAQ, being a zero-shot neural link prediction applicable to any KG, we re-used the checkpoint provided by the authors. Hyperparameters of all models are in App. D.2.

Fig. 3 reports the results of the selected baselines on the new benchmarks versus the old benchmarks. The results on the new benchmark show consistency w.r.t the number of reasoning steps needed to

Table 5: **A simple baseline as CQD outperforms most CQA SoTA on the new benchmarks** in terms of test MRR for different complex queries on FB15k237+H, NELL995+H and ICEWS18+H.

| | Model | 1p | 2p | 3p | 2i | 3i | 1p2i | 2i1p | 2u | 2u1p | 4p | 4i |
|---|---|---|---|---|---|---|---|---|---|---|---|---|
| FB15k237+H | GNN-QE | 42.8 | **6.5** | **4.2** | 10.3 | 10.3 | 4.6 | 7.8 | 36.9 | 7.2 | **3.8** | 13.2 |
| | ULTRAQ | 40.6 | 5.5 | 3.7 | 8.2 | 7.9 | 4.1 | 7.9 | 33.8 | 5.1 | 3.4 | 10.8 |
| | CQD | 46.7 | 6.3 | 2.7 | **18.4** | **16.2** | **5.6** | **9.4** | 34.9 | 7.3 | 1.1 | **16.5** |
| | ConE | 41.8 | 5.7 | 3.9 | 16.8 | 13.9 | 4.0 | 6.6 | 25.9 | 5.3 | 3.3 | 13.1 |
| | QTO | **46.7** | 5.9 | 3.5 | 13.5 | 11.8 | 4.7 | 8.8 | **37.3** | **7.4** | 3.2 | 13.0 |
| NELL995+H | GNN-QE | 53.6 | 7.5 | 4.0 | 20.0 | 14.9 | 8.5 | 8.0 | **63.2** | 5.3 | 2.9 | 10.9 |
| | ULTRAQ | 38.9 | 5.7 | 3.3 | 15.0 | 10.0 | 6.6 | 5.8 | 39.5 | 5.5 | 2.8 | 6.6 |
| | CQD | **60.4** | **10.7** | 4.2 | 36.0 | 25.6 | **14.2** | 12.1 | 61.0 | **12.6** | 2.0 | 16.8 |
| | ConE | 53.1 | 9.0 | 5.4 | **39.1** | **29.8** | 8.5 | 9.1 | 52.2 | 6.9 | 4.3 | **19.4** |
| | QTO | 60.3 | 10.4 | 5.2 | 28.4 | 19.6 | 10.2 | **12.4** | 62.7 | 11.5 | **5.0** | 12.5 |
| ICEWS18+H | GNN-QE | 12.2 | 1.9 | 0.8 | 3.5 | 2.4 | 2.7 | 3.1 | **8.6** | 2.2 | 0.5 | **1.7** |
| | ULTRAQ | 6.3 | 1.2 | 0.9 | 2.9 | 1.5 | 2.2 | 1.1 | 4.2 | 1.0 | 0.7 | 0.8 |
| | CQD | **16.6** | **2.6** | **1.2** | **5.9** | **3.2** | **4.3** | **6.1** | 8.6 | **4.5** | **0.9** | 1.7 |
| | ConE | 3.5 | 1.2 | 0.8 | 1.0 | 0.2 | 0.7 | 1.2 | 1.3 | 0.8 | 0.5 | 0.1 |
| | QTO | **16.6** | 2.3 | 0.9 | 4.6 | 2.6 | 3.5 | 4.9 | 8.4 | 3.2 | 0.6 | 1.3 |

predict an answer. For example the MRR of the queries having type 3i is always lower than 2i, which is also lower than 1p. On the other hand, the performance of these query types for the old benchmarks did not follow this pattern, as 3i queries had in all cases better performance than 2i, as true full-inference 3i queries were comprising only 0.1% of all answers (Table 1).

Table 5 shows the results of the selected baselines on the new benchmarks, for all query types. Note that there is no need for stratification on our new benchmarks. Furthermore, we excluded our CQD-Hybrid baseline from this evaluation, as the usage of a hybrid model is pointless for full-inference $(q, t)$ pairs. The surprising result from Table 5 is that CQD, which is the simplest baseline, relying only on a pre-trained link predictor, outperforms newer and more sophisticated methods in most cases. Furthermore, we highlight how 2u queries have similar performance than 1p, and 2u1p similar to 2p, as expected from Sec. 3. Finally, we remark that ICEWS18+H is much harder than NELL995+H and FB15k237+H, across all query types, even 1p, thus raising the for neural link predictors as well. This confirms our hypothesis that the sampling method of the KG splits plays a big role in determining the hardness of the benchmark.

> **Takeaway 3.**
>
> Our benchmarks reflect the true "hardness" requirement imposed by query types and should be used to evaluate the model CQA performance on missing links. Additionally, ICEWS18+H highlights that more realistic sampling strategies are more challenging for current SoTA.

## 7 CONCLUSION

In this paper, we revisit CQA on KGs and reveal that the "good" performance of SoTA approaches predominantly comes from answers that can be reduced to easier types (Table 2), the vast majority of which boiling down to single link prediction (Table 1). We also propose an hybrid solver, CQD-Hybrid, that by combining classical graph matching approaches with neural query answering explicitly reduces query answer pairs to easier query types when existing links are available and therefore can surpass SoTA on old benchmarks. We then created a set of new benchmarks FB15k237+H,NELL995+H and ICEWS18+H that only consider full-inference queries and are much more challenging for current SoTA approaches. We consider them to be a stepping stone towards benchmarking truly complex reasoning with ML models.

However, both old and our new benchmarks only consider queries with bindings of single target variables. While this reflects an important class of queries on real-world KGs, many real-world queries require bindings to multiple target variables (i.e. answer tuples). We plan to extend our current study of the "real hardness" of CQA benchmarks to queries involving negation (Zhang et al., 2021), and to other popular settings in neural query answering, such as inductive scenarios (Galkin et al., 2022) where some entities and/or relations are unseen during the test stage.

## REPRODUCIBILITY STATEMENT

We report the hyperparameter settings of all compared models in App. D . Our code and new benchmarks are included as supplemental materials in an anonymized GitHub repo[5] and will be made available upon acceptance.

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

## A  ADDITIONAL ANALYSIS ON THE OLD BENCHMARKS

**Additional comparisons**  Table A.4 and Table A.3 show the full results of FB15k237 and NELL995, including the results comparing all methods, all query types, and the subtypes they are reduced to.

Moreover, A.2 show the full results of the union queries, including the old overall results, which includes $(q, t)$ pairs having non-existing links in their reasoning tree (see Fig. A.1), and the "new overall", in which those $(q, t)$ pairs are filtered out. In particular, for "up" queries, we observe that for FB15k237, the full-inference queries have a higher MRR than the overall, while for NELL995, we find the MRR results on full-inference $(q, t)$ pairs to be lower than the new overall results and then the partial inference $(q, t)$ pairs as expected. We attribute the results on FB15k237 to the presence of bias in the $(q, t)$ pairs due to their scarcity in the benchmarks. We argue that for the same reason 2u1p performances are not comparable to 2p. However, we will show that on the new benchmark Sec. 6 our claim is confirmed.

### A.1  INFLUENCE OF EXISTING LINKS ON QUERIES INVOLVING NEGATION

To show that our analysis can be valuable also for different kind of queries, in Table A.1 we show an analysis for queries involving negation. For such analysis, we split the reasoning tree, defined in Sec. 3, into two subparts, namely *positive reasoning tree*, composed by the non-negated triples, and the *negative reasoning tree*, only composed by the negated triples. In particular, in the same spirit of Table 1, by only looking at the positive reasoning tree, we report the percentage of $(q, a)$ pairs that can be reduced to easier types (partial-inference) and the one that cannot be reduced to a simpler type (full-inference). Furthermore, we argue that also the negative reasoning tree contains training triples, but how this propagates to performance is less clear than the positive part, as each method treats negation differently. Table A.1 shows that in both FB15k237 and NELL995, queries can be reduced to easier types, revealing that our analysis also extends to negated queries. The only exception is 2in where the positive reasoning tree consists of a single link, see App. B. Consequently, no reduction to partial inference is possible (see Sec. 3).

### A.2  PERFORMANCE ANALYSIS OF QTO

We include the stratified analysis of QTO for the old benchmarks in Tables A.3 and A.4. For a fair comparison, we remark that we used the same link predictor for CQD, CQD-Hybrid, and QTO. Details and hyperameters are available in App. D.1. Our analysis reveals that, similarly to the other

Table A.1: **Most negated queries have training links in the positive reasoning tree of the** $(q, a)$ **pairs**. Partial-inference and full-inference refer to the **positive** reasoning tree. '-' refers to reductions that are not possible.

| | FB15k237 | | NELL995 | |
|---|---|---|---|---|
| | partial-inference | full-inference | partial-inference | full-inference |
| 2in | - | 100 | - | 100 |
| 3in | 95.4 | 4.6 | 93.9 | 6.1 |
| 2pi1pn | 98.4 | 1.6 | 97.8 | 2.2 |
| 2pni1p | 100 | 0.0 | 100 | 0.0 |
| 2in1p | 97.5 | 2.5 | 95.9 | 4.1 |

Table A.2: **Full inference "2u"** $(q, t)$ **pairs show higher performance than non-filtered, while full-inference "up" pairs show lower or comparable performance.** Results on union queries for existing benchmarks and comparison between old and new overalls. '-' refers to reductions that are not possible, while '/' to reductions that are possible but that are not present in the data.

| Query type | method | FB15k237 | | | | | NELL995 | | | | |
|---|---|---|---|---|---|---|---|---|---|---|---|
| | | overall | overall (new) | 1p | 2u | 2u1p | overall | overall (new) | 1p | 2u | 2u1p |
| 2u | GNN-QE | 16.2 | **40.7** | / | **40.7** | - | 15.4 | 34.8 | / | 34.8 | - |
| | ULTRAQ | 13.2 | 33.6 | / | 33.6 | - | 10.2 | 21.3 | / | 21.3 | - |
| | CQD | **17.6** | 32.7 | / | 32.7 | - | 19.9 | 35.9 | / | 35.9 | - |
| | ConE | 14.3 | 29.9 | / | 29.9 | - | 14.9 | 28.2 | / | 28.2 | - |
| 2u1p | GNN-QE | **13.4** | 9.7 | 8.7 | 50.9 | 13.1 | 8.8 | 8.8 | 6.3 | 53.1 | 2.0 |
| | ULTRAQ | 10.2 | 7.3 | 6.6 | 33.2 | **15.0** | 8.4 | 7.0 | 6.6 | 37.1 | 6.5 |
| | CQD | 11.3 | 10.5 | 10.4 | 13.7 | 14.7 | 16.0 | 14.6 | 14.2 | 51.8 | 9.5 |
| | ConE | 10.6 | 7.4 | 7.0 | 20.3 | 11.9 | 11.1 | 7.0 | 9.4 | 43.2 | 4.0 |

baselines, QTO performance drop when evaluated on the full-inference $(q, a)$ pairs only (diagonal), w.r.t overall results. Moreover, even when QTO is the SoTA on a certain query type, most of the time it is not so on the portion of full-inference $(q, a)$ pairs only, showing that improving performance on the partial-inference $(q, a)$ pairs not necessarily results on improvements over the full-inference ones.

Results on NELL995, shown in Table A.3, reveal that CQD-Hybrid outperforms QTO for some query types of NELL995, i.e. 3i, 1p2i, 2i1p, suggesting that even by only setting a score=1 to the training triples it is possible to obtain SoTA results on the old benchmarks. We remark that, while both in CQD-Hybrid and QTO a score=1 is set to the training links, QTO is much more sophisticated than CQD-Hybrid, as they 1) calibrated the scores with a heuristics, 2) store a matrix $|V| \times |V|$ for each relation name containing the score for every possible triples, 3) have a forward/backward mechanism in the reasoning.

**Influence of intermediate existing entities on the results** For query types having intermediate variables, such as "2p", "3p", "pi", "ip", and "up", we analyzed the number of existing intermediate entities matching the intermediate variables (existentially quantified variables) could influence the results. We refer to this number as "**cardinality of existing entities**". For example, in Fig. 1 the entities "When in Rome" and "Spiderman 2" are existing intermediate entities for the query $q_1$. The intuition is that while the presence of some of these entities might simplify the query answering Sec. 3, at test time we do not know which entity leads to a reduction and which does not, as we do not have access to the test data. Hence, as we claim that the existing models memorize the training data Sec. 5, we argue that a high value of cardinality must lead to lower performance, as those entities would act as noise for the models, hindering its capability of predicting the correct answers higher in the ranking.

To support our claim, we analyze the percentage of $(q, t)$ pairs having different values of cardinality, namely "0", "1", from "2" to "9", from "10" to "99", more than 100 ("100+"), shown in Fig. A.4 both for partial-inference and full-inference $(q, t)$ pairs, and their MRR, using CQD (Arakelyan et al., 2021), shown in Fig. A.3. Both Figs. A.3 and A.4 have bar charts for partial-inference (blue), and full-inference (orange) $(q, t)$ pairs for the difference cardinality categories, Fig. A.3 shows that CQD

Table A.3: **The best model on all available queries (column "overall") and the best on the full inference queries (diagonal) is different for 6/7 query types**, excluding 1p and 2u, those (q,a) pairs being all of type full-inference. Comparison of MRR scores for SOTA methods for the different sub-type queries of NELL995 can be reduced to. Best in bold.

| Query type | method | overall | 1p | 2p | 3p | 2i | 3i | 1p2i | 2i1p | 2u | 2u1p |
|---|---|---|---|---|---|---|---|---|---|---|---|
| 1p | GNN-QE | 53.6 | 53.6 | - | - | - | - | - | - | - | - |
|  | ULTRA-query | 38.9 | 38.9 | - | - | - | - | - | - | - | - |
|  | ConE | 60.0 | 60.0 | - | - | - | - | - | - | - | - |
|  | CQD | **60.4** | **60.4** | - | - | - | - | - | - | - | - |
|  | CQD-hybrid | **60.4** | **60.4** | - | - | - | - | - | - | - | - |
|  | QTO | 60.3 | 60.3 | - | - | - | - | - | - | - | - |
| 2p | GNN-QE | 17.9 | 18.2 | 6.1 | - | - | - | - | - | - | - |
|  | ULTRA-query | 11.2 | 11.5 | 4.6 | - | - | - | - | - | - | - |
|  | ConE | 16.0 | 16.3 | 7.2 | - | - | - | - | - | - | - |
|  | CQD | 22.0 | 22.3 | **7.6** | - | - | - | - | - | - | - |
|  | CQD-hybrid | 23.8 | 24.2 | 6.2 | - | - | - | - | - | - | - |
|  | QTO | **24.7** | 25.1 | 7.3 | - | - | - | - | - | - | - |
| 3p | GNN-QE | 15.2 | 15.1 | 8.1 | 3.4 | - | - | - | - | - | - |
|  | ULTRA-query | 9.7 | 9.8 | 5.1 | 4.1 | - | - | - | - | - | - |
|  | ConE | 13.8 | 13.8 | 8.2 | 6.5 | - | - | - | - | - | - |
|  | CQD | 13.4 | 12.8 | 7.8 | **8.5** | - | - | - | - | - | - |
|  | CQD-hybrid | 17.8 | 17.2 | 9.9 | 8.1 | - | - | - | - | - | - |
|  | QTO | **22.6** | 22.3 | 10.9 | 7.7 | - | - | - | - | - | - |
| 2i | GNN-QE | 40.0 | 41.4 | - | - | 3.6 | - | - | - | - | - |
|  | ULTRA-query | 36.3 | 37.5 | - | - | 3.2 | - | - | - | - | - |
|  | ConE | 39.6 | 40.6 | - | - | 8.0 | - | - | - | - | - |
|  | CQD | 42.2 | 43.3 | - | - | **6.9** | - | - | - | - | - |
|  | CQD-hybrid | **44.2** | 45.7 | - | - | 6.7 | - | - | - | - | - |
|  | QTO | **44.2** | 45.7 | - | - | 5.1 | - | - | - | - | - |
| 3i | GNN-QE | 50.9 | 53.6 | - | - | 10.9 | 2.2 | - | - | - | - |
|  | ULTRA-query | 47.7 | 50.0 | - | - | 11.3 | 1.5 | - | - | - | - |
|  | ConE | 50.2 | 51.9 | - | - | 18.3 | 4.8 | - | - | - | - |
|  | CQD | 51.8 | 53.9 | - | - | 14.4 | **3.6** | - | - | - | - |
|  | CQD-hybrid | **57.8** | 61.6 | - | - | 10.7 | 2.5 | - | - | - | - |
|  | QTO | 56.2 | 60.0 | - | - | - | 2.4 | - | - | - | - |
| 1p2i | GNN-QE | 29.1 | 42.7 | 21.4 | - | 12.0 | - | 10.2 | - | - | - |
|  | ULTRA-query | 25.1 | 39.0 | 25.6 | - | 8.2 | - | 7.0 | - | - | - |
|  | ConE | 26.1 | 38.5 | 25.2 | - | 10.5 | - | 8.6 | - | - | - |
|  | CQD | 31.5 | 44.0 | 25.7 | - | 14.5 | - | **12.6** | - | - | - |
|  | CQD-hybrid | **33.2** | 48.7 | 24.2 | - | 13.7 | - | 9.4 | - | - | - |
|  | QTO | 32.8 | 47.4 | 22.0 | - | 14.5 | - | 9.8 | - | - | - |
| 2i1p | GNN-QE | 20.5 | 20.8 | 16.3 | - | 14.6 | - | - | 20.7 | - | - |
|  | ULTRA-query | 15.6 | 16.5 | 9.5 | - | 9.6 | - | - | 11.4 | - | - |
|  | ConE | 17.6 | 17.7 | 16.4 | - | 25.7 | - | - | 19.5 | - | - |
|  | CQD | 25.8 | 25.9 | 21.2 | - | 26.3 | - | - | 23.6 | - | - |
|  | CQD-hybrid | **28.4** | 28.9 | 20.4 | - | 22.8 | - | - | 23.3 | - | - |
|  | QTO | 28.2 | 28.5 | 20.2 | - | 24.0 | - | - | **24.4** | - | - |
| 2u | GNN-QE | 34.8 | - | - | - | - | - | - | - | 34.8 | - |
|  | ULTRA-query | 21.3 | - | - | - | - | - | - | - | 21.3 | - |
|  | ConE | 28.2 | - | - | - | - | - | - | - | 28.2 | - |
|  | CQD | 35.9 | - | - | - | - | - | - | - | 35.9 | - |
|  | CQD-hybrid | 35.9 | - | - | - | - | - | - | - | 35.9 | - |
|  | QTO | **37.6** | - | - | - | - | - | - | - | **37.6** | - |
| 2u1p | GNN-QE | 8.8 | 6.3 | - | - | - | - | - | - | 53.1 | 2.0 |
|  | ULTRA-query | 8.4 | 6.6 | - | - | - | - | - | - | 37.1 | 6.5 |
|  | ConE | 7.0 | 9.4 | - | - | - | - | - | - | 43.2 | 4.0 |
|  | CQD | 14.6 | 14.2 | - | - | - | - | - | - | 51.8 | 9.5 |
|  | CQD-hybrid | 15.0 | 14.5 | - | - | - | - | - | - | 53.5 | **9.6** |
|  | QTO | **16.4** | 15.8 | - | - | - | - | - | - | 61.2 | **9.6** |

is highly influenced by the value of cardinality having decreasing performance at the increase of the cardinality. Note that by matching this plot with the one in Fig. A.4, one can understand if the result on a specific query type are highly influenced by a high proportion of $(q, t)$ pairs having the same cardinally. For example, the MRR of 2u1p full-inference $(q, t)$ pairs in FB15k-237 is highly influenced by the presence of about 50% of $(q, t)$ pairs having cardinality of "0".

This additional analysis shows that there is an additional level of "hardness" that can be considered when designing a benchmark or evaluating a model.

**Imbalances of relation names and anchor nodes**   Next, we check if there is some imbalances in both relation names and anchors. Note that if a relation name or an anchor entity is present multiple

Table A.4: **The best model on all available queries (column "overall") and the best on the full inference queries (diagonal) is different for 5/7 query types**, excluding 1p and 2u, those (q,a) pairs being all of type full-inference. Comparison of MRR scores for SOTA methods for the different sub-type queries of FB15k237 can be reduced to. Best in bold.

| Query type | method | overall | 1p | 2p | 3p | 2i | 3i | 1p2i | 2i1p | 2u | 2u1p |
|---|---|---|---|---|---|---|---|---|---|---|---|
| 1p | GNN-QE | 42.8 | 42.8 | - | - | - | - | - | - | - | - |
|  | ULTRA-query | 40.6 | 40.6 | - | - | - | - | - | - | - | - |
|  | ConE | 41.8 | 41.8 | - | - | - | - | - | - | - | - |
|  | CQD | **46.7** | **46.7** | - | - | - | - | - | - | - | - |
|  | CQD-hybrid | **46.7** | **46.7** | - | - | - | - | - | - | - | - |
|  | QTO | **46.7** | **46.7** | - | - | - | - | - | - | - | - |
| 2p | GNN-QE | 14.7 | 14.8 | 4.7 | - | - | - | - | - | - | - |
|  | ULTRA-query | 11.5 | 11.5 | 4.2 | - | - | - | - | - | - | - |
|  | ConE | 12.8 | 12.8 | **5.2** | - | - | - | - | - | - | - |
|  | CQD | 13.2 | 13.3 | 3.5 | - | - | - | - | - | - | - |
|  | CQD-hybrid | 15.0 | 15.2 | 3.5 | - | - | - | - | - | - | - |
|  | QTO | **16.6** | 16.7 | 4.0 | - | - | - | - | - | - | - |
| 3p | GNN-QE | 11.8 | 12.0 | 4.4 | 4.8 | - | - | - | - | - | - |
|  | ULTRA-query | 8.9 | 9.0 | 4.6 | 4.4 | - | - | - | - | - | - |
|  | ConE | 11.0 | 11.0 | 5.4 | 2.6 | - | - | - | - | - | - |
|  | CQD | 7.8 | 7.8 | 3.4 | 1.8 | - | - | - | - | - | - |
|  | CQD-hybrid | 11.0 | 11.1 | 3.6 | 4.4 | - | - | - | - | - | - |
|  | QTO | **15.6** | 15.8 | 4.5 | **5.0** | - | - | - | - | - | - |
| 2i | GNN-QE | 38.3 | 39.3 | - | - | 4.0 | - | - | - | - | - |
|  | ULTRA-query | 35.7 | 36.7 | - | - | 3.4 | - | - | - | - | - |
|  | ConE | 32.6 | 33.3 | - | - | 5.5 | - | - | - | - | - |
|  | CQD | 35.0 | 35.8 | - | - | **7.3** | - | - | - | - | - |
|  | CQD-hybrid | 37.6 | 38.7 | - | - | 3.8 | - | - | - | - | - |
|  | QTO | **39.7** | 40.8 | - | - | 5.7 | - | - | - | - | - |
| 3i | GNN-QE | 54.1 | 56.0 | - | - | 10.9 | 5.2 | - | - | - | - |
|  | ULTRA-query | 51.0 | 52.9 | - | - | 9.7 | 4.3 | - | - | - | - |
|  | ConE | 47.3 | 48.3 | - | - | 15.6 | 4.0 | - | - | - | - |
|  | CQD | 48.5 | 49.6 | - | - | 17.0 | **6.0** | - | - | - | - |
|  | CQD-hybrid | 52.7 | 54.8 | - | - | 9.9 | 4.6 | - | - | - | - |
|  | QTO | **54.6** | 56.4 | - | - | 15.4 | 5.4 | - | - | - | - |
| 1p2i | GNN-QE | 31.1 | 32.8 | 15.5 | - | 5.7 | - | 9.1 | - | - | - |
|  | ULTRA-query | 29.6 | 31.5 | 19.2 | - | 4.2 | - | 7.4 | - | - | - |
|  | ConE | 25.5 | 26.6 | 13.9 | - | 5.4 | - | **9.7** | - | - | - |
|  | CQD | 27.5 | 28.7 | 13.4 | - | 7.1 | - | 9.0 | - | - | - |
|  | CQD-hybrid | 31.2 | 33.4 | 14.8 | - | 4.8 | - | 7.0 | - | - | - |
|  | QTO | **33.8** | 35.9 | 15.8 | - | 6.2 | - | 7.3 | - | - | - |
| 2i1p | GNN-QE | 18.9 | 19.2 | 8.2 | - | 6.2 | - | - | 3.4 | - | - |
|  | ULTRA-query | 18.6 | 18.9 | 8.5 | - | 5.1 | - | - | **8.1** | - | - |
|  | ConE | 14.0 | 13.8 | 9.7 | - | 12.8 | - | - | 5.6 | - | - |
|  | CQD | 20.7 | 21.0 | 10.5 | - | 6.7 | - | - | 7.6 | - | - |
|  | CQD-hybrid | 24.0 | 24.6 | 10.0 | - | 4.6 | - | - | 7.5 | - | - |
|  | QTO | **24.7** | 25.3 | 10.3 | - | 8.6 | - | - | **8.1** | - | - |
| 2u | GNN-QE | **40.7** | - | - | - | - | - | - | - | **40.7** | - |
|  | ULTRA-query | 33.6 | - | - | - | - | - | - | - | 33.6 | - |
|  | ConE | 29.9 | - | - | - | - | - | - | - | 29.9 | - |
|  | CQD | 32.7 | - | - | - | - | - | - | - | 32.7 | - |
|  | CQD-hybrid | 32.7 | - | - | - | - | - | - | - | 32.7 | - |
|  | QTO | 37.0 | - | - | - | - | - | - | - | 37.0 | - |
| 2u1p | GNN-QE | 9.7 | 8.7 | - | - | - | - | - | - | 50.9 | 13.1 |
|  | ULTRA-query | 7.3 | 6.6 | - | - | - | - | - | - | 33.2 | **15.0** |
|  | ConE | 7.4 | 7.0 | - | - | - | - | - | - | 20.3 | 11.9 |
|  | CQD | 10.5 | 10.4 | - | - | - | - | - | - | 13.7 | 14.7 |
|  | CQD-hybrid | 10.3 | 10.0 | - | - | - | - | - | - | 18.5 | 13.6 |
|  | QTO | **12.2** | 11.6 | - | - | - | - | - | - | 35.3 | 11.3 |

times in a query, this is counted only once. As shown in Table A.5, for FB15k237 and NELL995 we find that in most cases there is no predominant relation name nor anchor entity. However, there are exceptions, as for 3i queries of FB15k237, where the anchor node "USA" is present in 30.1% of them, and for *2u* queries, where it is present in 22.5%. We also notice that there is a predominance of USA as an anchor entity across the vast majority of query types, which is most likely given by the vast presence of this entity in the knowledge graph.

Figure A.3: **Higher cardinality of intermediate existing entities leads to lower MRR.** Influence of the number of existing intermediate variables on the MRR for datasets FB15k237 and NELL995, using CQD.

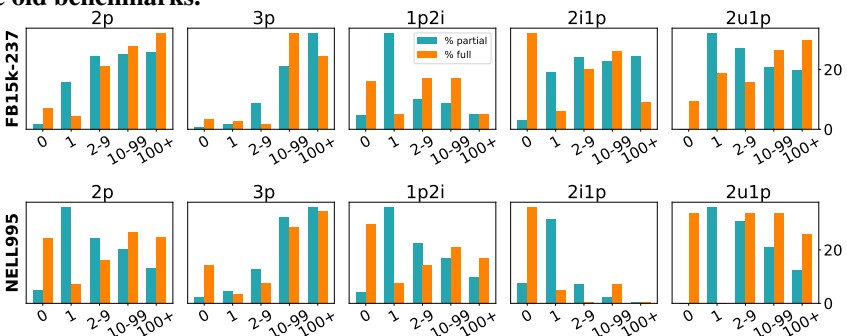

Figure A.4: **Proportion of $(q, t)$ pairs having different cardinality of intermediate existing entities in the old benchmarks.**

Table A.5: **There is no significant unbalancement in anchor nor relation names for each query type** . Most frequent relation name and anchor per query type in the old benchmarks. Highest value in bold.

| Query | relation name | frequency(%) | anchor entity | frequency(%) |
|---|---|---|---|---|
| | | FB15k237 | | |
| 1p | film_release_region$^{-1}$ | 3.7 | USA | 0.7 |
| 2p | location/location/contains | 7.8 | USA | 4.0 |
| 3p | location/location/contains | 10.5 | USA | 3.3 |
| 2i | people/person/profession$^{-1}$ | 16.3 | USA | 16.4 |
| 3i | people/person/gender$^{-1}$ | **18.7** | USA | **30.1** |
| pi | people/person/gender$^{-1}$ | 13.4 | USA | 14.1 |
| ip | people/person/profession$^{-1}$ | 7.8 | US dollars | 5.7 |
| 2u | film/film/language$^{-1}$ | 14.6 | USA | 22.5 |
| up | taxonomy_entry/taxonomy$^{-1}$ | 9.9 | /m/08mbj5d | 5.4 |
| | | NELL995 | | |
| 1p | concept:atdate | 6.6 | concept_stateorprovince_california | 0.5 |
| 2p | concept:proxyfor$^{-1}$ | 5.5 | concept_lake_new | 1.5 |
| 3p | concept:proxyfor$^{-1}$ | 11.9 | concept_book_new | 2.5 |
| 2i | concept:atdate$^{-1}$ | **21.6** | concept_book_new | 7.6 |
| 3i | concept:atdate$^{-1}$ | 21.1 | concept_company_pbs | **12.0** |
| pi | concept:proxyfor$^{-1}$ | 9.6 | concept_book_new | 6.1 |
| ip | concept:subpartof$^{-1}$ | 7.0 | concept_athlete_sinorice_moss | 3.8 |
| 2u | concept:atdate$^{-1}$ | 10.4 | concept_company_pbs | 4.9 |
| up | concept:subpartof$^{-1}$ | 8.5 | concept_athlete_sinorice_moss | 3.3 |

# B  QUERY TYPES INCLUDING NEGATION

In Fig. A.2 we include the query types including the negation operator involved in the analysis described in App. A.1. Those queries are obtained by adding a negation to a *single* triple pattern of some of the query structures described in Sec. 2. For example, by negating a triple pattern involved

in the intersection of a 2i, 3i, 2i1p queries, we obtain respectively:

$$?T : (a_1, r_1, T) \land \neg(a_2, r_2, T), \tag{2in}$$

$$?T : (a_1, r_1, T) \land (a_2, r_2, T) \land \neg(a_3, r_3, T), \tag{3in}$$

$$?T : \exists V_1.((a_1, r_1, V_1) \land \neg(a_2, r_2, V_1) \land (V_1, r_3, T)), \tag{2in1p}$$

Moreover, by placing the negation in different triple patterns of the query type 1p2i, we obtain two query types:

$$?T : \exists V_1.((a_1, r_1, V_1) \land (V_1, r_2, T) \land \neg(a_2, r_3, T)), \tag{2pi1pn}$$

where the negation is placed in the triple pattern directly involved in the intersection, and

$$?T : \exists V_1.((a_1, r_1, V_1) \land \neg(V_1, r_2, T) \land (a_2, r_3, T)), \tag{2pni1p}$$

where the negation is placed on a triple pattern involved in the path query 2p.

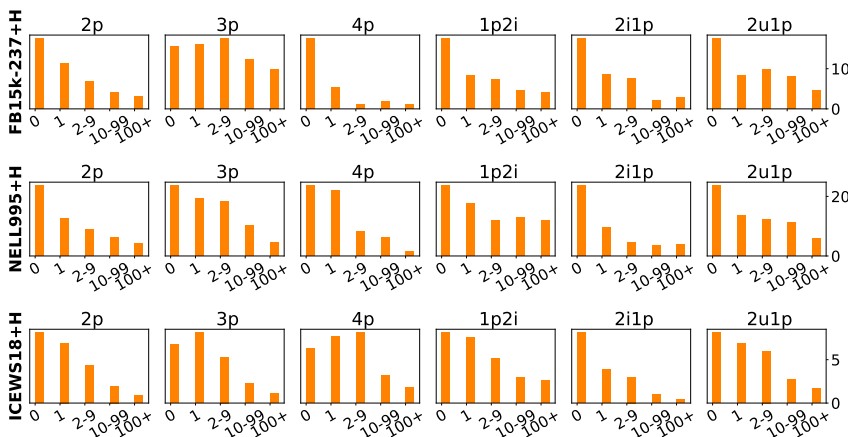

Figure C.1: **Higher cardinality of intermediate existing entities leads to lower MRR.** Influence of the number of existing intermediate variables on the MRR for datasets FB15k237+H, NELL995+H, ICEWS18+H, using CQD.

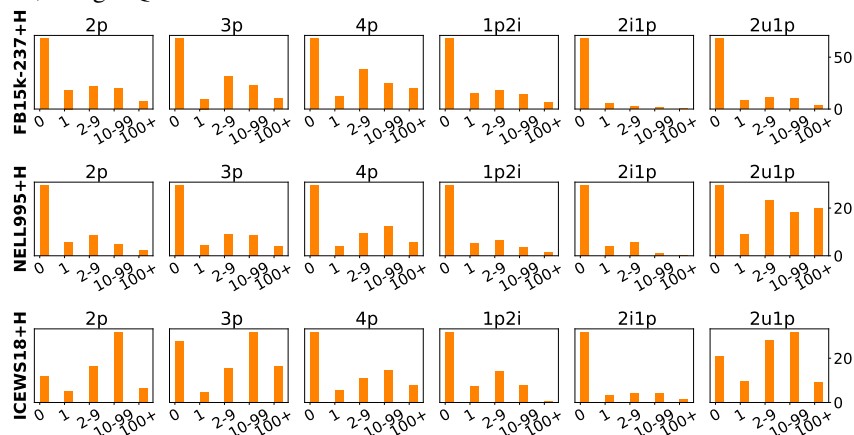

Figure C.2: **Proportion of** $(q, t)$ **pairs having different cardinality of intermediate existing entities in new benchmarks.**

## C ADDITIONAL ANALYSIS ON THE NEW BENCHMARKS

**Influence of intermediate existing entities on the results**   Similar to what was done for the old benchmarks in App. A, we analyze the proportion and influence (on the MRR) of the number of intermediate existing entities for CQD, on the new benchmarks FB15k237+H, NELL995+H, ICEWS18+H. Fig. C.1 show a very similar result to the one in Fig. A.3, showing a decreasing MRR at the increase of the value of cardinality for all benchmarks. On the other hand the proportions of the full-inference $(q, t)$ pairs on the new benchmarks Fig. C.2 are quite different than the one shown in Fig. A.4, having in general an higher number of $(q, t)$ pairs with cardinality "0". Nevertheless, the overall MRR on those query types on the new benchmark is much lower than the one on the old Fig. 3, showing that the "hardness" of a $(q, t)$ pair is mainly defined by (1) number of hops/number of reasoning steps, (2) being of type partial-inference or full-inference.

| Query | relation name | frequency(%) | anchor entity | frequency(%) |
|---|---|---|---|---|
| | | FB15k237+H | | |
| 1p | film_release_region$^{-1}$ | 3.7 | USA | 0.7 |
| 2p | marriage/type_of_union$^{-1}$ | 19.7 | American english | 1.1 |
| 3p | people/person/nationality$^{-1}$ | 14.4 | Hartford | 1.2 |
| 2i | film_release_region$^{-1}$ | 10.2 | USA | 4.6 |
| 3i | people/person/profession$^{-1}$ | 11.8 | USA | 9.2 |
| pi | people/marriage/type_of_union$^{-1}$ | 19.6 | USA | **20.0** |
| ip | webpage/category$^{-1}$ | **20.0** | /m/02s9vc | 3.5 |
| 2u | film_release_region$^{-1}$ | 10.4 | USA | 4.7 |
| up | webpage/category$^{-1}$ | **20.0** | /m/02s9vc | 4.9 |
| 4p | people/person/profession$^{-1}$ | 19.3 | Hartford | 3.0 |
| 4i | people/person/profession$^{-1}$ | 18.6 | USA | 19.4 |
| | | NELL995+H | | |
| 1p | concept:atdate | 6.6 | concept_stateorprovince_california | 0.5 |
| 2p | concept:proxyfor$^{-1}$ | 16.3 | concept_book_things | 0.6 |
| 3p | concept:proxyfor$^{-1}$ | 19.8 | concept_lake_new | 0.7 |
| 2i | concept:atdate$^{-1}$ | 10.2 | concept_lake_new | 1.9 |
| 3i | concept:atdate$^{-1}$ | 15.1 | concept_lake_new | 3.7 |
| pi | concept:proxyfor$^{-1}$ | 18.7 | concept_book_new | **17.4** |
| ip | concept:proxyfor$^{-1}$ | 19.4 | concept_lake_new | 9.7 |
| 2u | concept:atdate$^{-1}$ | 10.1 | concept_lake_new | 1.9 |
| up | concept:proxyfor$^{-1}$ | 18.3 | concept_lake_new | 6.8 |
| 4p | concept:proxyfor$^{-1}$ | **20.0** | concept_stateorprovince_colorado | 1.5 |
| 4i | concept:atdate$^{-1}$ | **20.0** | concept_date_n2004 | 8.0 |
| | | ICEWS18+H | | |
| 1p | Make statement$^{-1}$ | 11.12 | USA | 3.0 |
| 2p | Make Statement$^{-1}$ | 19.9 | United Stated | 3.4 |
| 3p | Make Statement$^{-1}$ | 19.9 | Saudi Arabia | 2.7 |
| 2i | Consult$^{-1}$ | 16.4 | Citizen (India) | 3.2 |
| 3i | Consult$^{-1}$ | **20.0** | Turkey | 5.5 |
| pi | Make statement$^{-1}$ | 18.8 | United States | 7.4 |
| ip | Make statement$^{-1}$ | 19.9 | Saudi Arabia | 5.3 |
| 2u | Consult$^{-1}$ | 16.5 | Citizien (India) | 3.1 |
| up | Make statement$^{-1}$ | **20.0** | Saudi Arabia | 5.4 |
| 4p | Make statement | 19.8 | Saudi Arabia | 2.9 |
| 4i | Consult | **20.0** | Turkey | **8.4** |

Table C.1: **The new benchmarks are generated such that anchors and relation names cannot appear more than 20% in each query type.** Most frequent relation name and anchor per query type in the new benchmarks.

**Imbalances of relation names and anchor nodes** When creating the new benchmarks FB15k237+H, NELL995+H, ICEWS18+H, to make sure that no anchor entities nor relation name was predominant in each query type, we set a maximum of 20% frequency for both anchor entities and relation names accross all benchmarks and query type, as shown in Table C.1

| Dataset | 2p | | 3p | | 2i | | 3i | | 1p2i | | 2i1p | | 2u | | up | |
|---------|-----|----|-----|----|-----|----|-----|----|-----|----|-----|----|-----|----|-----|----|
| | k | tn | k | tn | k | tn | k | tn | k | tn | k | tn | k | tn | k | tn |
| FB15k237 | 512 | p | 8 | p | 128 | p | 128 | p | 256 | p | 64 | p | 512 | m | 512 | m |
| NELL995 | 512 | p | 2 | p | 2 | p | 2 | p | 256 | p | 256 | p | 512 | m | 512 | m |

Table D.1: CQD hyperparameters old benchmarks. tn = tnorm. p=prod, m=min. Note that for 1p neither the k nor the tnorm are needed.

# D  HYPERPARAMETERS

In this section we detail the hyperparameters used for each dataset and model.

## D.1  OLD BENCHMARKS

**GNN-QE**  We did not tune hyperparameters, but re-used the ones provided in the official repo. [6]

**ULTRA**  We did not tune hyperparameters, but re-used the ones provided in the official repo.[7]

**CQD**  In our experiments we use ComplEx as it is a simple, robust and versatile predictor (Ruffinelli et al., 2020). Hence, we re-used the pre-trained link predictor (Trouillon et al., 2017) provided by the authors.[8]  However, we tuned CQD-specific hyperparameters, namely the CQD beam "k", ranging from [2,512] and the t-norm type being "prod" or "min" In Table D.1 we provide the hyperparameter selection for the old benchmarks FB15k237 and NELL995. Also note that we normalize scores with min-max normalization.

**ConE**  We did not tune hyperparameters but re-used the ones provided in the official repo. [9]

**CQD-Hybrid**  For CQD-Hybrid, to make the comparison with CQD fair, we re-used the hyperparameters found for CQD and fixed an upper bound value for the CQD beam "k" to 512, even when there are more existing entities matching the existentially quantified variables, to match the upper bound of the "k" used for CQD. Additionally, the scores of the pre-trained link predictor are normalized between $[0, 0.9]$ using min-max normalization, and a score of 1 is assigned to the existing triples.

**QTO**  We re-used the hyperparameters provided in the official repo. [10] For a fair comparison with CQD, we used the same pre-trained link predictor for both methods.

## D.2  NEW BENCHMARKS

For FB15k237+H and NELL995+H, we re-used the same models trained for the old benchmarks, and using the same hyperparameters presented in App. D.1. Instead, for the new benchmark ICEWS18+H we trained every model. The used hyperparameters for each model are presented in the following:

**GNN-QE**  For GNN-QE, we tuned the following hyperparameters: (1) *batchsize*, with values 8 or 48, and *concat hidden* being True or False, while the rest are the same used for the old benchmarks and do not change across benchmarks. For ICEWS18+H the best hyperparameters are "batchsize=48" and "concat hidden=True".

**ULTRAQ**  Being a zero-shot neural link predictor, we re-used the same checkpoint provided in the official repo, as for D.1.

---

[6]https://github.com/DeepGraphLearning/GNN-QE/tree/master/config
[7]https://github.com/DeepGraphLearning/ULTRA/tree/main/config/ultraquery
[8]https://github.com/Blidge/KGReasoning/
[9]https://github.com/MIRALab-USTC/QE-ConE/blob/main/scripts.sh
[10]https://github.com/bys0318/QTO/tree/main

| Dataset | 2p | | 3p | | 2i | | 3i | | 1p2i | | 2i1p | | 2u | | up | | 4i | | 4p | |
|---------|---|----|---|----|---|----|---|----|-----|----|-----|----|---|----|---|----|---|----|---|----|
| | k | tn | k | tn | k | tn | k | tn | k | tn | k | tn | k | tn | k | tn | k | tn | k | tn |
| ICEWS18+H | 32 | p | 2 | p | 2 | p | 2 | p | 512 | p | 256 | p | 2 | m | 2 | m | 2 | p | 2 | p |

Table D.2: CQD hyperparameters new ICEWS18+H benchmark. tn = tnorm. p=prod, m=min

**CQD**   For NELL995+H and FB15k237+H we re-used the pre-trained link-predictor and the same hyperparameters found for the old benchmarks D.1. Instead, for the newly created ICEWS18+H, we train ComplEx Trouillon et al. (2017) link predictor with hyperparameters "regweight" 0.1 or 0.01, and batch size 1000 or 2000, with the best being, respectively 0.1 and 1000. Moreover, in D.2 are shown the hyperparameters for CQD on the new benchmark ICEWS18+H.

**ConE**   We re-used the same hyperparameters of NELL995 for ICEWS18+H.

**QTO**   The same considerations made in App. D.1 apply.

# E   KNOWLEDGE GRAPHS STATISTICS

The statistics, i.e., number of entities, relation names, training/validation/test links, of the knowledge graphs used in this paper are shown in Table E.1.

Table E.1: Statistics of knowledge graphs used to generate complex queries

| Dataset | Entities | Relation Names | Training Links | Validation Links | Test Links | Total Links |
|---------|----------|----------------|----------------|------------------|------------|-------------|
| FB15k237 | 14,505 | 237 | 272,115 | 17,526 | 20,438 | 310,079 |
| NELL995 | 63,361 | 200 | 114,213 | 14,324 | 14,267 | 142,804 |
| ICEWS18+CQ (*new*) | 20,840 | 250 | 213,304 | 25,048 | 24,689 | 263,041 |

