# OpenReview forum: "Is Complex Query Answering Really Complex?"
_ICLR.cc/2025/Conference — Submitted to ICLR 2025_

### Official Review · Reviewer_DNKW · 2024-10-28

**Soundness:** 3
**Presentation:** 3
**Contribution:** 2
**Rating:** 5
**Confidence:** 4

**Summary:**

In this paper, authors re-examine the existing problems of knowledge graph complex reasoning datasets. The authors propose that the current dataset cannot effectively measure the generalization ability of the reasoning model, that is, the complex queries in the dataset can be solved by the triples leaked in the training graph, and verifies their conjecture through extensive and sufficient experiments. Further, the authors propose a new set of benchmarks to more effectively measure the performance of complex reasoning models.

**Strengths:**

* Motivation of the paper is novel, and the re-examination of existing benchmarks is valuable.
* Experiments in this paper can support the conclusion well.
* Writing of the paper is good, the structure is clear, the layout is good, and it is easy to follow.

**Weaknesses:**

* Lack of discussion of related work, if the space is limited, this part can be placed in the appendix.
* In Section 5.1, the author proposes a CQD-Hybrid solver. Actually, the practice described in the paper is very similar to the QTO [1] and I think the difference should be cited and discussed.
* As an effort to propose new benchmarks, the experiments for the new benchmark are a little less. More baselines, some case analysis, etc., should be added.
* Some typos, such as line.468: 50.000

[1] Answering Complex Logical Queries on Knowledge Graphs via Query Computation Tree Optimization. In ICML2023

**Questions:**

* The problem of information leakage in training graphs can be solved well by the inductive setting in naive knowledge graph reasoning (one-hop reasoning) task. Actually, there have been some attempts to establish inductive settings in CQA[1][2], where there will be no information leakage because the training and test graphs are different. How do you think this paper differs from these works?
* In my opinion, link leaks in the training graph only affect the GNN based and neural link predictor based methods, while the embedding-based methods do not take advantage of the information in the training graph (except for 1p queries). Why does this type of approach also degrade on the new benchmark?
* As mentioned in weakness, what's the difference between CQD-Hybrid and QTO?


[1] Inductive Logical Query Answering in Knowledge Graphs. In NeruIPS 2022.

[2] Type-aware Embeddings for Multi-Hop Reasoning over Knowledge Graphs. In IJCAI 2022.

---

> ### Author Response · Authors · 2024-11-20
>
> We thank the reviewer for praising our motivation, contribution and presentation. We will address all the questions in the following. We believe we can do this easily and hope to reach full acceptance.
>
> >Lack of discussion of related work; CQD-Hybrid is very similar to the QTO [1] and I think the difference should be cited and discussed; difference between CQD-Hybrid and QTO?
>
> We added a discussion of related works in the revised version of the paper (page 8), including a paragraph of hybrid solvers like QTO, highlighting the differences with CQD-Hybrid in Appendix A.2.
> See also our global answer.
>
> > As an effort to propose new benchmarks, the experiments for the new benchmark are a little less. More baselines, some case analysis, etc., should be added.
>
>
> We added QTO as a baseline for the new benchmarks in the revised version of the paper. We are happy to add more baselines and run further analysis, if you could be more specific.
>
> >The problem of information leakage in training graphs can be solved well by the inductive setting in naive knowledge graph reasoning (one-hop reasoning) task.
>
>
> We agree, but we remark that the transductive setting we address is widely adopted and complementary to the inductive scenario. Therefore, investigating the issue of transductive benchmarks and fixing them without changing setting is an important contribution.
>
> That said, the fact that inductive benchmarks can be “more challenging” will depend on how they are created.  One would need to perform an in-depth analysis to the scripts and process that generate them as we did for the transductive. As in our case, the issue is not the transductive scenario, but the way the benchmarks have been generated and “blindly” used so far.
>
> >Actually, there have been some attempts to establish inductive settings in CQA[1][2], where there will be no information leakage because the training and test graphs are different. How do you think this paper differs from these works?
>
> We remark that we do not propose a benchmark for the inductive setting, rather propose a benchmark on the transductive setting that truly evaluates reasoning performance of transductive models on performing complex query answering. We will cite those works as related works, but clarifying they are addressing two different problems. One can have a rigorous transductive benchmark with no leakage, as we shown.
>
> >In my opinion, link leaks in the training graph only affect the GNN based and neural link predictor based methods, while the embedding-based methods do not take advantage of the information in the training graph (except for 1p queries). Why does this type of approach also degrade on the new benchmark?
>
>  Our experiments suggest that also the embedding-based methods are affected by this leakage. Due to their extensive training on several query types (they train on up to 150.000 of queries for 10 query types) we suspect that they implicitly take advantage of existing triples by memorizing them in the embeddings as they are trying to reconstruct the triple tensor [3]
>
> [3] Trouillon, Théo, et al. "Complex embeddings for simple link prediction." International conference on machine learning. PMLR, 2016.

---

> > ### Author Response · Authors · 2024-11-26
> >
> > We kindly ask the reviewer to review our response and our changes in the revised submission. We completed all the experiments we presented in the previous response, showing that even the performance of QTO drop when evaluated on the full-inference query-answer pairs of the old benchmark (diagonal of Tables A.3, A.4), and on the new benchmark (Table 5), with its performance being similar to CQD, a much simpler and older baseline.

---

> > > ### Comment · Reviewer_DNKW · 2024-12-02
> > >
> > > Thanks for your reply. However, considering the limitations of the motivational statement of this paper (the comparison between inductive settings and the link leaks in transductive settings) and some missing discussions with some related work, I think this paper needs a major revision, so I tend to maintain the score.

---

> > > > ### Author Response · Authors · 2024-12-02
> > > >
> > > > >considering the limitations of the motivational statement of this paper (the comparison between inductive settings and the link leaks in transductive settings)
> > > >
> > > > Which motivational statement are you referring to? We do not see the point, as transductive and inductive are two distinct settings that are evaluated separately.
> > > > We remark that we do not propose a benchmark for the inductive setting, rather propose a benchmark on the transductive setting that truly evaluates reasoning performance of transductive models on performing complex query answering.
> > > >
> > > > > some missing discussions with some related work
> > > >
> > > > We added a discussion of QTO in Appendix A.2, along with experiments of QTO for the old benchmarks, Tables A.3,A.4 and the new benchmark in Table 5. We remark that we do not consider FIT as it is equivalent to QTO on the query types we considered (see appendix G.2, and Table 5 of FIT). Could you provide specific references for the related works that are missing?

---

### Official Review · Reviewer_z2gA · 2024-11-03

**Soundness:** 2
**Presentation:** 3
**Contribution:** 2
**Rating:** 5
**Confidence:** 4

**Summary:**

The hard answers studied in complex logical queries are those that cannot be retrieved due to gaps in the knowledge graph (KG). This paper reclassifies these hard answers into two categories: full-inference and partial-inference The authors argue that partial inference can be reduced into simpler query types and partial inference occupies the majority of  existing datasets, BetaE. They discover that current models perform poorly on full inference tasks and propose a new benchmark to highlight this issue. Additionally, they introduce a new model specifically designed to tackle partial inference answers by explicitly retrieving existing links from the training knowledge graph.

**Strengths:**

1. This paper find a interesting weakness of existing CQA dataset and propose a useful method and benchmark.
2. This paper is well written and easy to follow.

**Weaknesses:**

1. The baselines lack of the symbolic methods like QTO and FIT, which are the mainstream of CQA methods. The used CQD is a old symbolic method.
2. BetaE have three KGs but only two KGs are presented in the paper.
3.  The argument of 'reduced to easier types' is weird because query types with less constraint will be easy to solved than original query types, for example the performance of 3i is good than 2i. I suggest the authors use a preciser expression.
4. I disagree your arguments that your proposed CQD-hybrid is the first an hybrid solver. QTO and FIT use the information from observed KG and trained link predictor to construct the matrix and can use the hybrid information of train edges and pre-training embeddings.
5. Because of Weak 4, I am curious that the performance of symbolic method QTO and FIT as they already have the hybrid information.

**Questions:**

1. Do you vary your argument in train queries? I am wondering  the phenomenon that existed CQA models fails is caused by the train datasets have too many partial inference answers. Thus I am curious about the performance of symbolic search methods where these methods don not use queries to train.

---

> ### Author Response · Authors · 2024-11-20
>
> We thank the reviewer for their comments and for deeming our work well written, easy to follow and highlighting an important issue.
>
> > The baselines lack of the symbolic methods like QTO and FIT, which are the mainstream of CQA methods;  I am curious that the performance of symbolic method QTO and FIT as they already have the hybrid information
>
> We thank the reviewer for pointing out the existence of such symbolic methods. In the revised version of the paper, we will also add a comparison with other neuro-symbolic methods like QTO. We do not consider FIT as it is equivalent to QTO on the query types we considered (see appendix G.2, and Table 5 of FIT). See our global answer concerning QTO reporting new results. We are in the process of adding complete results for the new benchmarks. The bottom line is that QTO performance is comparable to the other solvers: that is the new benchmarks reveal the “true hardness” of CQA. Our story and contribution still holds.
>
> >BetaE have three KGs but only two KGs are presented in the paper.
>
> As we mentioned in the paper “ We do not consider FB15k [1] as it suffers from data leakage [2]”, we do not use FB15k as it was found to suffer from a data leakage problem [2].
>
> [1] Bordes, Antoine, et al. "Translating embeddings for modeling multi-relational data." Advances in neural information processing systems 26 (2013).
>
> [2] Toutanova, Kristina, and Danqi Chen. "Observed versus latent features for knowledge base and text inference." Proceedings of the 3rd workshop on continuous vector space models and their compositionality. 2015.
>
> > 'reduced to easier types' is weird because query types with less constraint will be easy to solved than original query types, for example the performance of 3i is good than 2i.
>
> The performance of 3i being much better than 2i is an indication that the old benchmarks are not good for evaluating true complex query answering performance. In fact, when adding another constraint there should be a drop in performance, as we’re adding an additional prediction that carries an error.
> Instead, in the old benchmarks there is a significant increase in performance because such added links were already present in the training data, making the query easier, not harder!
>
>
>
> >CQD-hybrid is not the first an hybrid solver. QTO and FIT use the information from observed KG and trained link predictor
>
> We thank the reviewer for pointing out the existence of such symbolic methods. We amended the line “To the best of our knowledge, this is the first time such an hybrid solver is proposed in the CQA literature.” and added more lines to contextualize QTO on page 8. We remark that CQD-Hybrid is not the main contribution of the paper, see our global answer above.
>
>
>
> >  Do you vary your argument in train queries? I am wondering the phenomenon that existed CQA models fails is caused by the train datasets have too many partial inference answers. Thus I am curious about the performance of symbolic search methods where these methods don not use queries to train.
>
> No, we re-used the same training queries. Which “arguments” are you exactly referring to?

---

> > ### Comment · Reviewer_z2gA · 2024-11-25
> > **Other concerns**
> >
> > Thank you for your hard work in addressing my concerns. However, the experiments over QTO are strange, and the performance is even worse than ConE and CQD in certain query types. It does not make sense that the symbolic method has such poor performance.

---

> > > ### Author Response · Authors · 2024-11-26
> > > **Follow up response**
> > >
> > > We first remark that, as specified in the Appendix D, we use the same link predictor for CQD, CQD-Hybrid and QTO for a fair comparison. Then, we simply run the code as available in the QTO repository. We provide the scripts and models to re-run our experiments in the anonymous repository here: https://github.com/anonsubmission7818/test-QTO. *We encourage you to reproduce our results and let us know if you cannot*.
> > >
> > > Analyzing the results, we stress that the drop in performance on the ***new benchmarks*** is expected. In fact, for full-inference query-answer pairs, hybrid solvers as QTO (but also CQD-Hybrid) have no advantage over other baselines since no training triple is in their reasoning tree.
> > >
> > > Instead, if you refer to the ***old benchmarks***, for partial-inference queries, QTO is the SoTA in most cases, as shown in non-diagonal cells of Table A.3, A.4. The performance drop you might refer to comes from using the pre-trained link predictor we used in our experiments.
> > >
> > > As you mention that we addressed all your previous concerns, we now hope that no one is missing and that the score will be updated accordingly. Please let us know if something is still not clear.

---

> ### Comment · Reviewer_z2gA · 2024-11-28
> **Reproduce issues**
>
> Thank you for providing the anonymous repository to reproduce the experimental results.
>
> However, I encountered some issues while trying to run your experiments. First, several necessary files are missing, including **dataset.py** and the **src/ directory**, as well as the **benchmark data**. After obtaining the necessary files and data from the QTO repository and your anonymous repository referenced in the main paper, I still faced a data loading issue, specifically:  **Exception has occurred: UnpicklingError invalid load key, 'v'**.

---

> > ### Author Response · Authors · 2024-11-28
> > **Files re-upload**
> >
> > Thank you very much for the effort in trying to reproduce our results. The error might have been related to the .pkl files in the repository – hence, we re-uploaded them here: https://github.com/anonsubmission7818/test-QTO/releases/tag/v.1.0.1. Also, please note that you have to recompute the adjacency matrix using our pre-trained models for a fair comparison. To ensure that, it’s enough to remove any precomputed .pt files in QTO-main (“rm -f QTO-main/*.pt”), and the adjacency matrices will be recomputed automatically.
> >
> > Please let us know if this solves the issue! In case of additional problems, please let us know, and please try to provide as many details about the issue as possible, such as the terminal commands and stack traces, so that we can resolve it in a timely manner.

---

### Official Review · Reviewer_fPSh · 2024-11-03

**Soundness:** 3
**Presentation:** 3
**Contribution:** 1
**Rating:** 5
**Confidence:** 5

**Summary:**

This paper studies the complex query answering task on knowledge graph and questions whether the data in existing dataset is unqualified. To be specific, the author proposes to term those pairs of complex query & answers which the corresponding reasoning process can leverage some parts of the knowledge in the training graph as partial inference pair and thus evaluate existing CQA models on full-inference pairs. This paper conducts extensive experiments to showcase this observation and analysis on some certain query types like 2u.

**Strengths:**

1. The key observation of this paper is interesting. The partial-inference pair is prevailing in existing datasets and the paper shows that full-inference pair is empirically much harder than partial-inference pair and thus the reasoning ability of SOTA CQA models may be less powerful than our imagination.

2. This paper's case study and deep analysis are praiseworthy. For example, the paper studies the query type with union and additionally finds that if we filter out such pairs that can be accessed by just one link, the performance of 2u will increase significantly, similar to that of the 1p query type.

**Weaknesses:**

1. Firstly, the discussion of the query type is constrained in this paper. Most dominantly, almost all researches conducted on complex query answering in recent years included negative queries yet this paper avoids that completely. Perhaps it's a drawback of their model design originating from the initial CQD paper, or perhaps the reasoning process defined in this paper fails in a negative query. Either way, it's problematic as the scope of the query type it investigated is strictly contained.

2. The claim of SOTA CQA models fail significantly on so-called full-inference pair is questionable, as it doesn't include recent models that are built by symbolic search algorithms, like QTO[1] and FIT[2], which use neural link predictors combined with searching algorithms and seems to bypass the challenges proposed by full-inference pair.  As the paper itself proposes a symbolic search method, the missing baselines in other symbolic search methods is questionable.

**Questions:**

The comparison of 2u-filter is dubious. As the definition of union query just requires one link to hold in the graph, I do not see the necessity to do such filtering as Figure A.1 as it more resembles 2i query type after filtering.

---

> ### Author Response · Authors · 2024-11-20
>
> We thank the reviewer for considering our paper interesting and praising our deep analysis. We proceed by answering their questions, believing all can be easily addressed.
>
> > almost all researches conducted on complex query answering in recent years included negative queries
>
> Note that, since queries with negation contain a “positive” sub-query and a ***single*** negative link, our analysis can be easily carried over as the positive sub-query can be reduced to a simpler type in the presence of training links.
> In fact, in FB15k-237 and NELL995, for queries involving negation, we analyzed the presence of existing links in the non-negative reasoning tree of the (q,a) pairs, and we can reveal that 95.4% of 3in query-answer pairs and 98.4% of pin (q,a) pairs in FB15k-237 have existing links present in the non-negative part of their reasoning tree. We will provide such percentages for the rest of the queries involving negation in the revised version of the paper in Table A.1 in Appendix A.1
>
> >SOTA CQA models fail significantly on so-called full-inference pair is questionable, as it doesn't include recent models that are built by symbolic search algorithms, like QTO[1] and FIT[2], which use neural link predictors combined with searching algorithms and seems to bypass the challenges proposed by full-inference pair
>
> In the revised version of the paper we will show that this is quite the opposite. Those hybrid models work great on the old benchmark, mainly because they’re constituted in the vast majority by partial-inference query-answer pairs, which QTO, FIT, and CQD-Hybrid exploit. In fact, when evaluating QTO on the portion of full-inference 2p queries of FB15k-237, the state-of-the-art remains ConE (table A.4), confirming how the perception of the progress in the field has been distorted due the the high presence of existing links. We do not consider FIT as it is equivalent to QTO on the query types we considered (see appendix G.2, and Table 5 of FIT).
> We are open to run more baselines if the reviewer has additional suggestions.
>
> > definition of union query just requires one link to hold in the graph, I do not see the necessity to do such filtering as Figure A.1 as it more resembles 2i query type after filtering.
>
>
> We do the filtering because some links are impossible.
> Even if it’s true that only one link needs to hold in the graph, union queries should evaluate how good CQA models are in performing such a union between two ***missing*** links. However, if only one link is missing and the other does not exist at all, we are measuring something different, which gives a ‘’false’’ sense of hardness of union queries: 2u queries should be as hard as 1p, not harder!
>
> In fact, 2u was reported to have an MRR much much lower than 1p, but this was not due to the hardness of the union itself, rather due to the presence of non-existing links in their reasoning tree. By filtering out such query-answer pairs we obtain performance that are similar to 1p.

---

> ### Comment · Reviewer_fPSh · 2024-11-26
>
> Dear Authors:
>     Thank you for making the rebuttals and I recognize your effort in adding QTO as one of your baselines to make the experiments more extensive.
>     However, I think the other concern is still valid.
>
> > We do the filtering because some links are impossible. Even if it’s true that only one link needs to hold in the graph, union queries should evaluate how good CQA models are in performing such a union between two missing links. However, if only one link is missing and the other does not exist at all, we are measuring something different, which gives a ‘’false’’ sense of hardness of union queries: 2u queries should be as hard as 1p, not harder!
>
> Regarding the discussion of the union query, if there must be two missing links, then what is the difference between a conjunctive query and a disjunctive query? 2u and 2i will be exactly the same! I think that violates the very definition of the disjunctive query.
>
> > Note that, since queries with negation contain a “positive” sub-query and a single negative link, our analysis can be easily carried over as the positive sub-query can be reduced to a simpler type in the presence of training links.
>
> ``that queries with negation contain a “positive” sub-query and a single negative link`` is only true based on the Q2B/BetaE dataset, it is far from something that can be taken as granted and can be wrong in more advanced dataset, therefore the whole discussion is really questionable when the query doesn't meet this condition and the definition of ``full-inference'' becomes dubious.

---

> ### Author Response · Authors · 2024-11-26
> **Follow up response**
>
> We thank the reviewer for recognizing that we addressed some of their concern, and we hope this can be reflected in a score update. We proceed by addressing the left concerns.
>
> > Regarding the discussion of the union query, if there must be two missing links, then what is the difference between a conjunctive query and a disjunctive query? 2u and 2i will be exactly the same! I think that violates the very definition of the disjunctive query.
>
> Let’s break this into the following points:
>
> 1) Overall, when answering a 2u union query, we’re interested in any link between the two. This is not the issue, however. The issue is that during ***evaluation*** in the old benchmarks, we have to ***deal with non-existing links***, that is, links that do not belong either to the train or test splits.  Evaluating a query-answer pair that has one non-existing link and one missing link in their reasoning tree is problematic. ***This is why we filter-out non-existing links and only evaluate on answers that have two missing links in their reasoning tree.***
>
> 2) Why is the presence of non-existing links problematic for evaluating 2u queries?
> Models are trained on existing links and non-existing links will have a very low probability to exist. Therefore, if at evaluation time we ask a model to perform well to non-existing links, the reported MRR will be lower than expected, as these links are out-of-distribution.
> This can be clearly seen in the old benchmarks, where 2u queries had a much lower MRR than 1p queries: on these benchmarks we are asking models to score triples that do not belong to the original KG.
>
> 3) Therefore we filter out non-existing links. Such filtering is done in the same spirit of filtering out easy answers in Q2B/BetaE datasets. By filtering out some answers we do not say they are *invalid*, rather that ***we do not evaluate on them***. While easy answers are filtered-out because they can be directly retrieved without predictions, answers that have non-existing links in their reasoning tree are filtered-out to evaluate the models’ performance in performing the union ***between missing links only.***
>
> 4) Even after the filtering, 2i and 2u are not the same, as ***still*** we’re evaluating two different operators. Logically, 2u is as hard as 1p, you should agree with this statement, because we just need the model to correctly predict one of the two missing links, while for 2i we still need both to hold. You can see that this was not reflected in the old benchmarks, but it is now reflected in the new benchmarks. 2u and 1p are roughly the same in terms of MRR and there is now a significant performance difference between 2u and 2i in the new benchmarks (Table 5).
>
> > **that queries with negation contain a “positive” sub-query and a single negative link** is only true based on the Q2B/BetaE dataset, it is far from something that can be taken as granted and can be wrong in more advanced dataset, therefore the whole discussion is really questionable when the query doesn't meet this condition and the definition of ``full-inference'' becomes dubious
>
> Which benchmarks are you referring to? The vast majority of CQA benchmarks that are commonly used, interpret negation queries in this way.
>
> We furthermore remark (again) that **regardless of the query type**, the analysis that we did can be extended to other benchmarks. Applying it to negation queries is just an example to show that this can be carried out in different ways to different query types. The only condition that we impose is that every link in the reasoning tree has to be only in the test graph; such condition can be applied to *any* query structure. However, if one wants to bring our analysis to other benchmarks, one need to check the way query are generated to make sure this condition is satisfied.
>
> We hope that we addressed all your concerns now, and we are happy to discuss it further if needed. We also encourage you to point to specific lines in the paper that might lead to a confusion.

---

### Official Review · Reviewer_z9wR · 2024-11-04

**Soundness:** 2
**Presentation:** 4
**Contribution:** 2
**Rating:** 3
**Confidence:** 4

**Summary:**

This manuscript presents a data-level study of knowledge graph complex query answering. The main argument is that the query-target pairs in existing datasets (q, t) can be somehow reduced to the easier ones (sub(q), t) if the required triple can be found in training KG. Therefore, the paper proposes to focus on the irreducible query answer pairs and empirically examine that the performance of all existing methods will drop significantly. The facts revealed above motivate a search approach highlighted by letting the edges in the train graph be memorized. The performance of the approach is compared against previous works on old and new benchmarks.

---

## Retrospective summary after the rebuttal period.

This author-reviewer discussion thread goes on for too long during the rebuttal period, and particularly, the later part of the debate becomes intense. I think it is necessary to provide such a summary to digest my concerns and how they are not fully addressed for future readers of this page. Finally, I respond to the authors' accusation of a goalpost shift.

### My concerns and why they are or are not addressed.

My initial concerns are, as stated in the weakness in the very first review:
1. the missing but essential baseline QTO/FIT.
2. the missing discussion in similar datasets, such as the BetaE dataset, FIT dataset, and EFO_k dataset.
3. distribution of the new benchmark.

The situation of those concerns are
1. Addressed.
2. Addressed partially. The BetaE dataset was considered empirically, which satisfies my minimum standard. However, no discussion about how their methodology of benchmark construction applies to query types in FIT and EFO_k datasets can be found.
3. It is still under debate, and let me expand on it as follows.

My concerns about the distribution of the proposed benchmark are decomposed into several fine-grained issues during the rebuttal period.

Distribution issues:
1. I proposed a new set of terminology that clearly describes two parts of samples $S_{X, II}$ and $S_{X, III}$ based on the split of training and testing edges. The old benchmark, although sampled in a synthetic data distribution (no matter how weird it is), still covers both sets. The new benchmark only focuses on the $S_{X, III}$. My view is that the ignorance of $S_{X, II}$ is worrisome because such ignorance failed to reflect CQA's performances on link prediction, logical connectives, and variables on $S_{X, II}$.
2. Besides, the new benchmark, even used jointly with the old ones, actually stresses the importance of $S_{X, III}$, which is picked by enforcing the missing links. By emphasizing such a subset of data, the author tries to encourage a better link prediction because the simplest way of eliminating the gap between the new benchmark and the old one is just to make a better link prediction.

Both two issues are not fully addressed.

1. The authors acknowledge the new benchmark's ignorance. However, they tried to alleviate the negative impact in two ways. The first way is the joint usage or the stratified analysis, which leads to my second issue. The second way is to state that $S_{X, II}$ is way less important than $S_{X, III}$, which leads to our disagreement in non-factual belief. My belief is the missing link (which needs to be filled to satisfy practical application) is relatively small and will be smaller due to the advancement of knowledge graph construction and knowledge harvesting. The authors believe that the missing links in a KG are significantly large proportion. Their belief is claimed to be supported by valid math derivation, which I don't think their math can support their belief.
2. For the second issue, I think the performance gap between old and new benchmarks is caused by only picking the missing links, and it can be narrowed by proposing better link predictors that have better link prediction performances on the missing links in existing KG datasets. There are some neverending debates about whether the gap can be fully closed or if it really reflects the measurement of CQA. It does not change my view that, as a benchmark that encourages researchers to get higher and higher scores, it is very important that a benchmark does not have a clear shortcut deviating from the original goal of the task (logical connectives and variables besides link prediction). Clearly, If one method can achieve better link prediction performance on test data, it can first predict all missing links and just run symbolic algorithms. The limit of this solution is exactly one begins with a link predictor that overfits the test set, which is how the authors create benchmark data. I don't think this is a valid outlook for constructing a new benchmark to facilitate the study. Ironically, optimizing on the old benchmark, criticized by the authors that are mostly measuring link prediction (suppose the authors are correct here), will finally close the gap between new and old benchmarks, making the new benchmark less and less useful.

### Some accusations from the authors.

The authors accused of continuous goalpost shifting. However, my concerns are centered on the data distribution of the new benchmark and how it will impact the study.

I decomposed and expanded my claims to respond to the authors' words, such as "we don't see any problems here.". I believe that the active authors deserve to know why I am against them. The authors, although very eloquent, repeatedly misunderstood my terms and words. Such as narrowing down my mentions of link predictors to neural link predictors (or knowledge graph embeddings) and refusing to accept my thoughts by just stating that neural models can never achieve perfect performance. However, no matter whether the neural models are perfect or not, the emphasis of new benchmarks on link prediction still remains unchanged.

I understand that people sometimes get emotional during the debate, and the emotional words from both the authors and me are already documented in the following threads. Please also let me express my apology if any of my words ever hurt anyone.

**Strengths:**

- This paper conducts an in-depth study of existing benchmarks and reveals biases regarding tree-shaped queries and union operators in several datasets.

**Weaknesses:**

I have two concerns about the content discussed and the angle studied in this paper.

Firstly, the content seems to be very old. I am not sure whether this paper has been recycled on a sufficiently long time, so the author is not aware of the recent progress in this field.
1. The dataset discussed only covers the query types in [1], which is outdated today. In 2024, several datasets covering far more complex queries are also proposed, including [2] for queries with logical negation, [3] for cyclic queries, and [4] for multi-answer variables. For the ``fair'' split of triples, the answer is also unaware of existing studies on temporal CQA [5].
2. The baselines discussed are also no later than the year 2023. ULTRAQ is almost the same as the GNN-QE.
3. Given the above ignorance, the proposed CQD-hybrid method is fundamentally identical to the QTO [6] proposed in ICML'23. Those two methods are all search-based approaches that involve memorizing the train edges, which is proposed in this paper and also reflected in Equation 4 in [6], noticing that normalizing link predictor scores into [0,0.9] will not change the order of solutions.

I prefer to recognize methodological identicality as unawareness rather than plagiarism. Therefore I didn't raise an ethical review flag.


Secondly, saying that "the existing benchmark is problematic" is questionable and somehow self-contradictory with this paper's philosophy of choosing outdated simple queries.
- On the one hand, scores on the previous benchmarks [1-5] are far from saturated because the average score is still less than 50 out of 100. Optimizing empirical models on previous benchmarks will also benefit the performance of the proposed "hard" benchmark. Meanwhile, recognizing the importance of training edges, although motivating the CQD-hybrid in this paper, is not new to the community because it is practiced in QTO [6] and later followed by FIT [3]. It hardly says why these findings are essential.
- On the other hand, the paper only focuses on the simpler query forms proposed in [1]. One might argue that such simple query forms cover a sufficiently large portion of real-world user cases, so the choice of such forms is reasonable. The same practical point of view can also apply to the easy-hard contrast produced by whether the reasoning triples of a query are observed or not. Although the previous benchmark consists of too many observed triples, as shown in this paper, it can also be reasonable by arguing that the train graph consists of a sufficiently large portion of knowledge that users are interested in.



References:

[1] Ren, H., Hu, W., & Leskovec, J. (2020). Query2box: Reasoning over knowledge graphs in vector space using box embeddings. arXiv preprint arXiv:2002.05969.

[2] Ren, H., & Leskovec, J. (2020). Beta embeddings for multi-hop logical reasoning in knowledge graphs. Advances in Neural Information Processing Systems, 33, 19716-19726.

[3] Yin, H., Wang, Z., & Song, Y. (2023). Rethinking Complex Queries on Knowledge Graphs with Neural Link Predictors. arXiv preprint arXiv:2304.07063.

[4] Yin, H., Wang, Z., Fei, W., & Song, Y. (2023). ${\rm EFO} _k $-CQA: Towards Knowledge Graph Complex Query Answering beyond Set Operation.

[5] Lin, X., Xu, C., Zhou, G., Luo, H., Hu, T., Su, F., ... & Sun, M. (2024). TFLEX: temporal feature-logic embedding framework for complex reasoning over temporal knowledge graph. Advances in Neural Information Processing Systems, 36.

[6] Bai, Y., Lv, X., Li, J., & Hou, L. (2023, July). Answering complex logical queries on knowledge graphs via query computation tree optimization. In International Conference on Machine Learning (pp. 1472-1491). PMLR.

**Questions:**

Please respond to my two concerns in the weakness part.

---

> ### Author Response · Authors · 2024-11-20
>
> We believe the main concerns of the reviewer can be fully addressed, as discussed in the general answer to all reviewers and in the single comments below.
>
> >The dataset discussed only covers the query types in [1], which is outdated today. In 2024, several datasets covering far more complex queries are also proposed, including [2] for queries with logical negation, [3] for cyclic queries, and [4] for multi-answer variables.
>
> We agree that more sophisticated queries are possible but i) the benchmarks we analyze are still widely used (see our global answer as well) and therefore our message to the community using them is valid and ii) more sophisticated benchmarks such as those using negation are still based on the same principle, and can be affected in the same way.
>
> See our global answer above: training links are also present in negation queries. We will update the paper with full results. For example, in 95.4% of 3in query-answer pairs and 98.4% of pin in FB15k-237 there are training links in the non-negative part of their reasoning tree. We report these values in Table A.1 in Appendix A.1.
>
>
>
> The bottom line is that regardless of how intricated a query structure is, the hardness of a query-answer pair depends on the number of training links that can be found in its reasoning tree
>
> >For the ``fair'' split of triples, the answer is also unaware of existing studies on temporal CQA [5]
>
>  We remark we are not solving temporal CQA. Instead, we use the temporal information to better sample classical triples (no temporal information is retained).
>
> Moreover, in [5] they use kg splits obtained by sampling uniformly at random the knowledge graph , while we create new splits s.t. the test only contains future triples (w.r.t the triples contained in train and valid).
>
> > baselines discussed are also no later than the year 2023
>
> That’s not true, as ULTRA-Query was published on Neurips 2024 (yet to be presented!). We acknowledge that QTO was not present, and we added to our revised paper. See our global answer where we show that also the performance of QTO depends on the data leak. In fact, if we perform our stratified analysis of QTO on FB15k-237,
>
>
> | Query type | all | 1p    | 2p   | 3p   | 2i   | 3i   | 1p2i | 2i1p | 2u   | 2u1p |
> |------------|---------|-------|------|------|------|------|------|------|------|------|
> | 1p         | 46.7    | 46.7  | -    | -    | -    | -    | -    | -    | -    | -    |
> | 2p         | 16.6    | 16.7  | 4.0  | -    | -    | -    | -    | -    | -    | -    |
> | 3p         | 15.6    | 15.8  | 4.5  | 5.0  | -    | -    | -    | -    | -    | -    |
> | 2i         | 39.7    | 40.8  | -    | -    | 5.7  | -    | -    | -    | -    | -    |
> | 3i         | 54.6    | 56.4  | -    | -    | 15.4 | 5.4  | -    | -    | -    | -    |
> | 1p2i       | 33.8    | 35.9  | 15.8 | -    | 6.2  | -    | 7.3  | -    | -    | -    |
> | 2i1p       | 24.7    | 25.3  | 10.3 | -    | 8.6  | -    | -    | 8.1  | -    | -    |
> | 2u         | 37.0    | -     | -    | -    | -    | -    | -    | -    | 37.0 | -    |
> | 2u1p       | 12.2    | 11.6  | -    | -    | -    | -    | -    | -    | 35.3 | 11.3  |
>
>
>
>
> it provides evidence that the aggregate performance reported in the papers is essentialy due to the ~98% of queries that can be reduced to 1p (see Table 1). For the full stratified analysis please refer to table A.3 and A.4.
> >the proposed CQD-hybrid method is fundamentally identical to the QTO [6] proposed in ICML'23. CQD-hybrid in this paper, is not new to the community because it is practiced in QTO [6] and later followed by FIT [3]. It hardly says why these findings are essential.
>
> We thank the reviewer for pointing out the existence of such hybrid solvers. We included it in our revision (page 8) and highlight how QTO performance is inflated (as CQD-Hybrid and all other solvers) by the current benchmarks.
>
> We remark that , CQD-Hybrid is definitely not identical to QTO. The only point in common between the two methods is setting a score=1 to the training links. However, QTO is much more sophisticated than CQD-Hybrid, as they 1) calibrate the scores with a heuristics, 2) store a matrix $|V|\times|V|$ for each relation containing the score for every possible triples, 3) have a forward/backward mechanism in the reasoning.
> CQD-Hybrid only set the score for the existing triples =1, proving that a pre-trained link predictor and memorization of the training triples *alone* are enough to obtain SoTA performance on the old benchmarks. We hence need a new benchmark where we cannot leverage existing links to truly evaluate a model’s reasoning capabilities and advance the field of complex query answering.

---

> > ### Author Response · Authors · 2024-11-20
> >
> > > the existing benchmark is problematic" is questionable and somehow self-contradictory with this paper's philosophy of choosing outdated simple queries
> >
> >
> > We state again that the issues we find in simple queries are carried over more complex queries. See our analysis for negative queries. What matters is the way the queries are created.
> >
> > > scores on the previous benchmarks [1-5] are far from saturated because the average score is still less than 50 out of 100.
> >
> > We did not claim scores are saturated, we claim that scores are inflated, in the sense that we are measuring the ability of a model to memorize training triples, not to reason at test time.
> > And it is definitely possible that models are not able to exactly memorize the whole training datasets (hybrid solvers can easily). This does not imply that they are not memorizing: they do! See our Tables.
> >
> > >Optimizing empirical models on previous benchmarks will also benefit the performance of the proposed "hard" benchmark.
> >
> >  This is not necessarily true. As shown in Table 2, and later on the new benchmark in Table 5, more sophisticated baselines not always perform better on the full-inference queries.
> > Even QTO fails short on the new benchmarks. For example, for FB15k-237+H :
> >
> > | Model        | 1p    | 2p    | 3p    | 2i    | 3i    | 1p2i  | 2i1p  | 2u    | 2u1p  |
> > |--------------|-------|-------|-------|-------|-------|-------|-------|-------|-------|
> > | GNN-QE   | 42.8 | **6.5** | **4.2** | 10.3  | 10.3  | 4.6 | 7.8   | 36.9  | 7.2 |
> > | ULTRAQ   | 40.6  | 5.5   | 3.7   | 8.2   | 7.9   | 4.1   | 7.9  | 33.8  | 5.1   |
> > | CQD      | **46.7** | 6.3 | 2.7   | **18.4** | **16.2** | **5.6** | **9.4** | 34.9 | 7.3   |
> > | ConE     | 41.8  | 5.7   | 3.9 | 16.8 | 13.9 | 4.0   | 6.6   | 25.9  | 5.3   |
> > | QTO      | **46.7** | 5.9   | 3.5   | 13.5  | 11.8  | 4.7   | 8.8   | **37.3** | **7.4** |
> >
> > For additional results on the new benchmark please see Table 5.
> >
> >
> > >Although the previous benchmark consists of too many observed triples, as shown in this paper, it can also be reasonable by arguing that the train graph consists of a sufficiently large portion of knowledge that users are interested in.
> >
> > This is true, and we argued for it by presenting a hybrid solver. The issue with the current benchmarks however is that: i) 98% of queries reduce to 1p only, which is unrealistic in the real-world and is a clear artifact of the creation of the benchmark 2) people do not realize that the performance they are reporting is essentially link prediction performance. We not only highlight these issues, but propose a new benchmark that is much more challenging and designed to avoid conflating memorization with reasoning.

---

> > > ### Comment · Reviewer_z9wR · 2024-11-25
> > > **Thanks for your contribution**
> > >
> > > Dear Authors,
> > >
> > > I acknowledge the efforts during the rebuttal period to add new baseline models (QTO) and new data (queries with logical negation). The empirical results obtained are certainly of high quality and with great detail. I also appreciate the author's efforts to populate new and hard benchmarks (C3).
> > >
> > > This paper's central claim is based on the fact that most of the samples in the old dataset are **partially inference queries**. It then claims that the scores from the old dataset are conflated because of memorization, so the scores are biased.
> > > However, I still hold another view for (C1) and (C2) even after carefully reading, comprehending, and reflecting on the quantitative results and claims. The disagreement is in how to interpret the results:
> > > - Firstly, it is too reckless to simplify reasoning as predicting new triples (inference-based).
> > >     - The definition of partial inference relies on the observation that removing one trivial edge in the reasoning tree can reduce the original query to a simpler query. This reduction is valid only when the logical calculus is conducted under boolean truth values and exhaustive search. It is **NOT** valid under a more realistic and machine-learning scenario, which is also suggested in Table A.3. I'm afraid I have to disagree with the differentiation of reasoning hardness by solely the categorization of full and partial inference queries.
> > >     - Table A.3 also supported my point. If the reduction is valid, one can predict the partial-inference 2p performance (2p-1p) with the full-inference 1p performance (1p-1p). However, the fact is that the relation is not even monotonic, see (GNN-QE 1p-1p < ConE 1p-1p but GNN-QE 2p-1p > ConE 2p-1p). This also happens for perfect memorization methods (CQD-hybrid 1p-1p > QTO 1p-1p but CQD-hybrid 2p-1p < QTO 2p-1p). The non-monotonic performances revealed that the performance of partial inference queries (even already reduced to 1p) is not dominated by the link prediction performance.
> > >   - I think this happens because different methods model the logical variables, quantifiers, and connectives in different ways. The various ways of parameterization/calculation make the actual implementation largely deviate from the **bipolar narrative of link memorization vs. link prediction suggested in this paper**. In other words, reasoning or query answering is more than link prediction.
> > > - Acknowledging that the reasoning is just more than predicting new links, the argument that **because 98% of queries reduce to 1p leads to a clear artifact** also becomes questionable. At least, this argument does not apply to the neural models without explicit memorization and boolean logical calculus because the reduction is not valid. Unfortunately, none of the evaluated baselines satisfy such a reduction.
> > >
> > > Best

---

> > > > ### Author Response · Authors · 2024-11-25
> > > > **Follow up response (part 1)**
> > > >
> > > > > I acknowledge the efforts during the rebuttal period to add new baseline models (QTO) and new data (queries with logical negation). The empirical results obtained are certainly of high quality and with great detail. I also appreciate the author's efforts to populate new and hard benchmarks (C3).
> > > >
> > > > We are happy **we answered most of your previous concerns and requests regarding experiments, types of queries used, baselines and writing**. We hope this can be reflected in a score change, as only one point is left open, concerning the “motivation”. We will address this below as we believe there is a clear misconception here that we can easily disentangle.
> > > >
> > > > It is easier if we start from your last comment.
> > > >
> > > > > the argument that because 98% of queries reduce to 1p leads to a clear artifact also becomes questionable. At least, this argument does not apply to the neural models without explicit memorization and boolean logical calculus because the reduction is not valid.
> > > >
> > > > Our claim is based on just statistics and we think you will agree: 98% of the queries of old benchmarks share a “syntactic” property, all their links but one are present in the training set. Even if you disagree on calling them “reducible to 1p”, you have to agree on the following points:
> > > >
> > > > A) this is a statistical property of the query distribution, we just counted number of links
> > > >
> > > > B) having 98%+ of this “type” of queries means that the overall performance – a simple arithmetic mean – will overcount the performance of this type more than other types
> > > >
> > > > C) this “type” (column “1p” in Tables 2, A.3, A.4) is incredibly **easier for all SOTA methods, regardless they are hybrid solvers or not**, than  the “other types” which are much harder (lower MRR) as empirical evidence shows (other columns of the previous Tables)
> > > >
> > > > D) the fact that “other types” are harder is not a distorted statistic coming from evaluation on few triples, because as we increase the number of queries (for which we need the new benchmarks) the MRR stays very low
> > > >
> > > > As such, from B+C+D it follows logically that the current benchmarks are skewed towards a certain “type” of queries that is inherently easier than the rest of the queries, and as we are computing arithmetic means. This inflates performance as the easier type is overrepresented. This reasoning is solidly based on numbers and we are not pushing any additional interpretation over this “type”. **This is enough to tell the community to be concerned with the current benchmarks** as we are measuring an aggregate performance over an overrepresented type.
> > > >
> > > > > Firstly, it is too reckless to simplify reasoning as predicting new triples (inference-based).
> > > >
> > > > This is a crucial misconception: We never claim one should simplify reasoning to predict *only* full-inference query. However, we need a full-inference only dataset to check point D above, and to thus set a more challenging bar. We elaborate more next.
> > > >
> > > > > This reduction is valid only when the logical calculus is conducted under boolean truth values and exhaustive search. It is NOTvalid under a more realistic and machine-learning scenario, which is also suggested in Table A.3.
> > > >
> > > >
> > > > We, in fact, say we do not want to discard the old benchmarks, but one has to use i) hybrid reasoners on them (**takeaway #2**) and ii) a stratified analysis on it (**takeaway #1**). Reporting aggregated means gives a misleading sense of performance. We agree with you that **reasoning in the real world needs hybrid reasoners**, but even with them, not doing a stratified analysis makes all performance collapse to the most represented, “easy type” only.

---

> > > > > ### Author Response · Authors · 2024-11-25
> > > > > **Follow up response (part 2)**
> > > > >
> > > > > > If the reduction is valid, one can predict the partial-inference 2p performance (2p-1p) with the full-inference 1p performance (1p-1p).
> > > > >
> > > > > This is not true, and we never claim it is true. **A reduction is a syntactic property of two query classes, it does not imply that a ML model should be equivalently good on both query classes**. The fact that a reasoner can benefit from it depends on how the reasoner is implemented. Note that **memorization in non-hybrid solvers is a spectrum**, and depends on the learning process and how the model is implemented. It is very well known that for a large embedding space, neural link predictors are able, in principle, to exactly reconstruct the original training tensor, see e.g., [7] and the literature of tensor factorizations and universal approximation of neural networks. However, for a small embedding size, and limited capacity, and given training dynamics, they can only partially reconstruct it in practice.
> > > > >
> > > > > This is expected whenever we are learning some ML model, there is *noise* in the optimization, and the learning problem is highly non-convex.
> > > > >
> > > > > This is why, from a clear ML perspective, predicting 2p->1p might not match exactly the performance of 1p->1p given the noise and the learning issues we mentioned above. Nevertheless, we do not need performance to match exactly to make our claim that the query type “2p->1p” is overrepresented and dominates the overall scores, see our points (A-D) above. Furthermore, the comparison with hybrid-solvers, who memorize exactly, brings further evidence that for non-hybrid models this happens for a certain kind of memorization (which again is not exact memorization, but some relaxation).
> > > > >
> > > > > [7] Trouillon, Théo, et al. "Complex embeddings for simple link prediction." International conference on machine learning. PMLR, 2016.
> > > > >
> > > > > >  The non-monotonic performances revealed that the performance of partial inference queries (even already reduced to 1p) is not dominated by the link prediction performance.
> > > > >
> > > > > We also do not need additional empirical evidence to say that this is a form of memorization happening: hybrid-solvers, which memorize exactly, score better by design on the old benchmarks, and much worse on the new benchmarks (see the performance drop of QTO). You cannot deny this empirical evidence.
> > > > >
> > > > > Note that *monotonicity is not needed* and is a confounder (that maybe is distracting you from our points A-D): monotonicity would only make sense if the models were not learning and there were no noise. As explained above, learning poses several challenges and we cannot match performance exactly. But there is already a clear statistical trend that can be measured: **performance drops when we move to harder classes**. This happens systematically (up to noise), and a regression plot is sufficient to see this.
> > > > >
> > > > >
> > > > > > The various ways of parameterization/calculation make the actual implementation largely deviate from the bipolar narrative of link memorization vs. link prediction suggested in this paper. In other words, reasoning or query answering is more than link prediction.
> > > > >
> > > > > This is essentially what we said above (and in the paper!). We invite you to re-read the paper with fresh eyes: we never claim an opposition between link memorization vs link prediction.
> > > > >
> > > > > If some specific line in the paper is still bothering you, please report it here and we are happy to discuss it further but also to amend it if needed, to make our point crystal clear.

---

> ### Comment · Reviewer_z9wR · 2024-11-26
> **Thanks for your reply.**
>
> Dear authors,
>
> Thanks for your point-to-point response. I would also like to engage more in the crucial discussion and skip some wording issues here and there. To summarize, I think we have already reached an agreement on the following observation:
>
> - The mean scores for almost all query types from existing benchmarks statistically **bury** the performance of full-inference query-target samples.
>
> And also, the following two arguments:
>
> - CQA is more than link memorization/prediction; it also includes the treatment of logical connectives and variables.
> - full-inference query-target pairs performed significantly worse than the average performance.
>
> Our major disagreement is rooted in how we understand such observations/arguments and what the implications are.
>
> - First, it is very important to understand why full-inference query-target pairs consist of a minor part of the query. Not surprisingly, this is the direct result of the fact that missing links (valid or test triples) are only a small ratio of all links in the investigated knowledge graph splits.
>    - Why are there roughly 98% percent of partial-inference query target pairs for 2p? The answer is simply because the missing links (valid+test links) comprise about 15%. Then, the probability of two links being missing could be roughly estimated as 15% * 15% = 2.25%. I think this estimation is valid since the ratio of non-reducible 2p in FB15k237 is 1.9% and 2.4% in NELL955.
>    - This means the full-inference query-target pairs become the significant **ONLY** when the ratio of missing links grows significantly. The ratio decays of full-inference queries will be less than 1% in many settings.
>
> If my understanding of the nature of the full-inference pair is correct, then several concerns prevent me from accepting the proposed settings as being valid.
> 1. **Are the full-inference query-target pairs truly complex?** No, they do not relate to the ``complex'' logical structure and do not align with the original goal of studying complex logical queries. Although this concept replacement makes the title, and other parts such as intro, section 6, and conclusion more eye-catching, it does not change the nature of sample manipulation. The difficulty of such query-target pairs, evident by the low MRR, is the result of selecting difficult samples from the original datasets, and the difficult sample is intuitively defined by how many edges in the queries are missing.
> 2. To me, the new benchmark follows another distribution in the probabilistic space of all possible query-target pairs. The new distribution is featured by all the reasoning edges being missing and, of course, has a smaller support than the previous data distribution. How small is the support of the new distribution? My previous estimation might provide some straightforward but rough intuition suggesting it is small, with a rough ratio of about $x^k$ where the $x\in[0, 1]$ is the ratio of the missing edges, $k$ is the number of the predicates in a specific query type.
> 3. Suggesting such a new benchmark sends readers important messages to pay more attention to this particular data distribution when optimizing their models. I am worried that the measurement from this distribution might be biased towards certain subsets of triples/entities (at least they are 10% missing triples). Then, **my question is whether this new data distribution remains valid and effective in evaluating the CQA method's comprehensive capability (for logical connectives and variables)**. From the description in the paper and the discussion before, I cannot see the answer.
> 5. Should we assign more attention to such particular data distribution from a developing point of view?  Also, following my understanding of the ratio of full-inference query-target pairs, my opinion is no. The reason is that the advancement of LLMs further reduces the hardness of knowledge harvesting. I conjecture that the ratio of missing links in KG will decay, so the ratio of full-inference pairs will also decay even faster.
>
> I still acknowledge the authors' job of identifying the more difficult part in the space of query-target pairs for a given query type.  Even though we separate this part of the samples from the averaging scores, it is still hard for people to select the most suitable model per query type with multiple benchmarks. Should they make decisions by averaging the one used before and the new one proposed here? If one is willing to choose a pure neural model, is it right for the model selection process to underweight the prediction of observed links?
>
> I am looking forward to your reply.
>
> Best

---

> > ### Author Response · Authors · 2024-11-27
> > **Follow up response (part 1)**
> >
> > We are happy that you are now agreeing with us on all points we make in the paper but one, we believe we have the perfect mathematical argument to solve it!
> >
> > > First, it is very important to understand why full-inference query-target pairs consist of a minor part of the query. Not surprisingly, this is the direct result of the fact that missing links (valid or test triples) are only a small ratio of all links in the investigated knowledge graph splits
> >
> > We believe that you fell prey of a logical fallacy: ***The fact that the current scripts generate queries in this way does not imply this is the best way nor the only possible way to do it.***
> >
> > First, consider that ***the number of missing links present in the benchmarks is arbitrary***, as it depends on the proportion of test triples selected in the original datasets (which were designed for link prediction many years ago). No one dictates that the test split should be 15%. Note that this is just a way to **simulate unknown links** (see our answer below, regarding missingness as unknowns).
> >
> > As this is arbitrary, try to perform this mental experiment: we could change this proportion, and you would see all the MRR results drop because more queries would be full-inference in the old benchmarks! In Machine learning, instead, changing the size of the test set should not change dramatically performance, because we assume i.i.d..
> > Again, these dataset splits were designed for 1p queries only, and used as-is, without much critical thinking for CQA.
> >
> > Second, ***the number of missing links does not necessarily imply that the number of full-inference queries is low***. In fact, we are able to get more than 50000 full-inference query instances for each type easily (*note we are not changing the train/val/test splits!*). We simply do not sample queries in the old way (that we discuss next).
> >
> > > Why are there roughly 98% percent of partial-inference query target pairs for 2p? The answer is simply because the missing links (valid+test links) comprise about 15%. Then, the probability of two links being missing could be roughly estimated as 15% * 15% = 2.25%. I think this estimation is valid since the ratio of non-reducible 2p in FB15k237 is 1.9% and 2.4% in NELL955
> >
> > The computation correctly computes the fraction of triples in the old benchmarks, but this is not representative of the “true” fraction of all possible queries. In fact, we can easily sample 50000 full-inference queries. Your wrong deduction assumes that one has to sample triples independently, but that is just an *assumption* and a non-realistic one. ***Triples are never independent in the real world***. It is however the simplest assumption one can make, and we conjecture it is in the old benchmarks because it was the simplest to implement.
> >
> > We not only get rid of that assumption, but also experiment with sampling queries according to a temporal pattern, which is more realistic.
> >
> >
> > > The difficulty of such query-target pairs, evident by the low MRR, is the result of selecting difficult samples from the original datasets, and the difficult sample is intuitively defined by how many edges in the queries are missing
> >
> > This is true, we want difficult samples, and we want to highlight that there is a correlation between the number of missing edges and low MRR, something that went unnoticed by the community and that you also seemed not to notice before our last message. We remark that ***difficult queries are not less frequent*** in the overall distribution.
> >
> > > This means the full-inference query-target pairs become the significant ONLY when the ratio of missing links grows significantly. The ratio decays of full-inference queries will be less than 1% in many settings.
> >
> > Again, the fact that the current ratio is 15% matters if you sample triples independently. We are questioning this way to sample triples, which biases the construction of the dataset towards easier partial-inference queries.
> > As argued above, we are not obliged to sample triples in this way, nor to keep the test split to 15% (we are keeping this for keeping the performance of 1p - link prediction).
> >
> >
> > > To me, the new benchmark follows another distribution in the probabilistic space of all possible query-target pairs. The new distribution is featured by all the reasoning edges being missing and, of course, has a smaller support than the previous data distribution. How small is the support of the new distribution? My previous estimation might provide some straightforward but rough intuition suggesting it is small, with a rough ratio of about  where the  is the ratio of the missing edges,  is the number of the predicates in a specific query type.
> >
> > Your estimation of the support is wrong, as you are sampling triples independently. The real support for full-inference queries is much larger, as we easily demonstrated in practice: we could flawlessly sample 50k full-inference query instances for all query types (and we can sample much more!)

---

> > > ### Author Response · Authors · 2024-11-27
> > > **Follow up response (part 2)**
> > >
> > > >Suggesting such a new benchmark sends readers important messages to outweigh this particular data distribution when optimizing their models.
> > >
> > > Rather than outweigh a particular data distribution a model can learn to reason without relying on training data. Would you rather continue to report performances that are clearly inflated due to memorization but claiming generalization instead? Note that the hybrid-reasoner class is tiny w.r.t. all the other classical machine learning models for CQA.
> > >
> > > > I am worried that the measurement from this distribution might be biased towards certain subsets of triples/entities (at least they are 10% missing triples).
> > >
> > > We already had such analysis on the new benchmark in Table C.1, which shows that there is no evident bias in the new benchmarks.
> > >
> > > > Then, my question is whether this new data distribution remains valid and effective in evaluating the CQA method's comprehensive capability (for logical connectives and variables).
> > >
> > > We remark again that this is not a new data distribution. We are simply getting rid of the simplifying assumption that triples are independent. We could turn the question the other way around: do you have any evidence that assuming triples are independent is more realistic?
> > >
> > > >Should we outweigh such particular data distribution from a developing point of view? Also, following my understanding of the ratio of full-inference query-target pairs, my opinion is no. The reason is that the advancement of LLMs further reduces the hardness of knowledge harvesting. I conjecture that the ratio of missing links in KG will decay, so the ratio of full-inference pairs will also decay even faster.
> > >
> > > We remark that presenting links to be inferred as “missing links“ in the sense that they could be known and people did not just put in sufficient effort, is a mistake. Links are missing not because people did not put enough effort (this happens, too), but because they cannot be known. In this context, they are not just "missing links" but rather "unknown missing links". These two things are not identical. We want to make inferences over unknown missing links. And we cannot assume that just more data in the future will solve the problem, this is wishful thinking.
> > >
> > > > it is still hard for people to select the most suitable model per query type with multiple benchmarks. Should they make decisions by averaging the one used before and the new one proposed here? If one is willing to choose a pure neural model, is it right for the model selection process to underweight the prediction of observed links?
> > >
> > > We believe you agreed that averages done blindly are evil and are distorting the perception of progress : )
> > > Averaging across the two datasets is problematic, and we discourage it. However, both benchmarks can still be used: the authors should ***report a stratified analysis on the old benchmarks*** and an aggregate analysis on the new benchmarks. We do not see a problem here. As we already stated in our takeaway messages.

---

> ### Comment · Reviewer_z9wR · 2024-11-30
> **Thanks for your further engagement.**
>
> Dear authors,
>
> I appreciate your patience and efforts in the discussion. I think this is what makes ICLR unique. With all due respect, I will try to make my response match the seriousness of yours :-).
>
>
> ## Recap of previous discussion
>
> In my first round of responses, and acknowledged by the follow-up feedback, we reached the following agreement.
> - **The full-inference query-target pairs / truly complex queries / irreducible query-answer pairs in your terminology are essentially hard samples.** The reasons are (1) the concept of query reduction does not apply to neural models and (2) treating hard samples as logically more complex queries is a concept replacement. You acknowledged those two points in your response. Then, the problem becomes how you blame the statistical average.
> - CQA is more than link memorization and prediction. **Treatment of other logical elements (connectives and variables) is equally important**.
>
> In my second round response:
> - I mention that even when you consider the hard samples and blame the statistical averaging, the ratio of hard queries is often small. Then, it is risky to construct a benchmark before we confirm whether the data is sampled from a distribution that can evaluate other logical elements with a **sufficiently wide coverage** (as we have already agreed on the importance of logical elements).
> After reading your response, you may have confused the concept of **ratio** with **frequency**. Your reasoning is that because one can get a good number of hard samples, one can say that the underlying distribution of all hard samples has a good coverage.
>
> ## The response in the third round
>
> In this round, I will first discuss the coverage of hard samples precisely. Then, I will express again why building such a benchmark with ONLY hard samples of this kind is not necessary.
>
> ### The coverage of hard samples.
>
> A knowledge graph contains the knowledge we observed, and we acknowledge there is also the knowledge that is missing. So in a knowledge graph dataset, the edges are split into two parts: the observed edges $E_{o}$ and the (artificially) missing edges $E_{m}$. In this part, we assume that the knowledge graph is given and the ratio of (artificially) missing edges is small.
>
> For a fixed query type, say 2p, the set of all possible query-target pairs is a fixed set $S_{2p}$ and the set of all reasoning trees is also another fixed set $T_{2p}$
> - Each $t\in T_{2p}$ is a 2-path. Then there are three categories for a 2-path: (I) all edges are observed; (II) part of edges are (artificially) missing, and the other edges are observed; (III) all edges are (artificially) missing. Resulting three subsets of $T_{2p}$, that is, $T_{2p,I}$, $T_{2p,II}$, and $T_{2p,III}$ of reasoning trees.
> -  Those subsets of reasoning trees directly relate to the query-answer pairs. And let's just use the same meaning of suffixes I, II, and III to justify the subset of query-target samples $S_{2p, I}, S_{2p, II}, S_{2p, III}$ due to its natural connection with the reasoning trees.
> - Notably, one query-answer pair can be explained by multiple reasoning trees, and if I understood the paper correctly, all reasoning trees $t$ for each full-inference query-target pair $s\in S_{2p, III}$ should belong to $T_{2p, III}$.
>
> What should we care about? For a fixed query type X, samples from $S_{X,I}$ are not evaluated in previous practice because it is already solved by an existing database system, samples from $S_{X,II}\cup S_{X,III}$ are then evaluated. And many methods solve $S_{X,II}\cup S_{X,III}$ as the goal.
>
> When $|E_m| << |E_o|$, it is natural that $|T_{X, III}| << |T_{X, II}|$ and then $|S_{X,III}| << |S_{X,II}|$. For 2p case, $|T_{2p, III}|$ is the number of 2-paths with edge $|E_m|$ but $|T_{2p, II}|$ is the number of 2-path with one edge from $E_m$ and another from $E_o$. What the paper has found in this paper reflects these kinds of differences. This fact is not affected by how you sample the data. It is about the range of data you can sample from.
>
> Constructing benchmarks from a distribution over $S_{X, III}$ ignores the entire $S_{X, II}$. The original manuscript argues that $S_{X,II}$ can actually be "reduced" to type III samples that belong to a simpler sub-query type $Y$. $S_{Y, III}$ is already measured, so we don't need to repeat that job in $S_{X,II}$. However, we all agree that the reduction is NOT valid, as I mentioned earlier. And as a result, the performance of logical connectives and variables on $S_{X,II}$ are NOT measured in your benchmark.
>
> Please be aware that the discussion above holds regardless of the sampling algorithm. Your previous responses about how you sample 50k samples in $S_{X,III}$ still do not contribute to any information in $S_{X,II}$.

---

> > ### Author Response · Authors · 2024-12-01
> > **You are assuming triple independence, why is this realistic? (I)**
> >
> > > With all due respect, I will try to make my response match the seriousness of yours :-).
> >
> > ***We believe that scientific debate should be based on rigorous logic and facts and the tone of the conversation should be polite***. We are trying to steer the conversation towards this direction, only partially succeeding so far. And trying to navigate the continuous goalpost shifting. We are glad that reviews and discussion are public.
> >
> > > The full-inference query-target pairs / truly complex queries / irreducible query-answer pairs in your terminology are essentially hard samples. The reasons are (1) the concept of query reduction does not apply to neural models and (2) treating hard samples as logically more complex queries is a concept replacement.
> >
> > We ***never agreed on these two points***. The concept of query reduction is model-independent, it is a syntactic property of query-answer pairs. ***The fact that “neural models” (i.e., non-hybrid models) are able to effectively memorize training triples is solid evidence in the community*** and spawn from the tensor literature and ML literature. Please read again the references we provided in our paper and answer above.
> >
> > The fact that you admit them as “hard sample” – *something many in the community ignore, including you in your review and previous answers dismissed it* – is already pointing to one of our contributions: **there is a curve of hardness, with “easy samples” being those reducible to 1p. There is solid evidence all models find them easier than the rest. And these samples constitute 98%+ of all datasets.** Current aggregate statistics are inflating perceived progress.
> >
> >
> > >  the ratio of hard queries is often small. [...] you may have confused the concept of ratio with frequency. Your reasoning is that because one can get a good number of hard samples, one can say that the underlying distribution of all hard samples has a good coverage.
> >
> > We believe the confusion appears on your side (and a frequency is a ratio!). We will try again to break the reasoning process step by step. We believe the flawed reasoning stems from the underlying flawed assumptions:
> > the current percentage of train/val/ test splits are highly representative of the real-world
> > missing triples are less than known triples
> > the sampling process is not flawed
> >
> >
> >
> >
> > > In this part, we assume that the knowledge graph is given and the ratio of (artificially) missing edges is small.
> >
> > This is the first flawed assumption.You cannot do this and base it only on the fact that current benchmarks have been split in a certain way.
> > *Why should we split the KG to have artificially/missing edges?* To estimate performance on unseen scenarios. As in ML, we would like this performance not to change with the split ratio much.
> >
> > First, we remark that benchmarks have been created for link predictions, then adapted to CQA. Second, consider that ***in a given KG, the number of real missing edges is far greater than the number of seen edges***. Consider just the proportion of edges in the current FB, NELL or even WikiData, where one can find millions of entities, this implies the possibility to have thousands of billions of possible triples. But only a fraction are observed. So in the real world $|E_m| >> |E_o|$. Note that this is true even when you discard those edges than might violate logical constraints in the ontology. Third, ***this only increases when you start counting complex queries beyond 1p***.
> >
> > If we want to measure CQA in real, challenging scenarios, we need to test our models under this perspective, not blindly inheriting benchmarks that were thought for simple link prediction.
> >
> > > For a fixed query type, say 2p, the set of all possible query-target pairs is a fixed set. [...] Resulting three subsets of $T_{2p}$, that is, $T_{2p,I}$, $T_{2p,II}$, and $T_{2p,III}$ of reasoning trees. Those subsets of reasoning trees directly relate to the query-answer pairs. And let's just use the same meaning of suffixes I, II, and III to justify the subset of query-target samples $S_{2p, I}, S_{2p, II}, S_{2p, III}$ due to its natural connection with the reasoning trees.
> >
> >
> > We appreciate ***you are explaining to us what we wrote in the paper about reductions***. There is no need to introduce new notation, let’s stick to the precise nomenclature we proposed.

---

> > > ### Author Response · Authors · 2024-12-01
> > > **You are assuming triple independence, why is this realistic? (II)**
> > >
> > > > When $|E_m| < |E_o|$, it is natural that $|T_{X, III}| << |T_{X, II}|$ and then $|S_{X,III}| << |S_{X,II}|$ [...] This fact does not affect how you sample the data. It is about the range of data you can sample from.
> > >
> > >
> > > You do not provide any rigorous reason why this should happened and unfortunately you cannot provide a “natural” derivation here unless you i) fix a starting ratio and ii) assume a sampling process.
> > >  ***The numerosity of query-answer pairs in a benchmark cannot be detached from the way you sample them***. You never rebutted on the fact that ***assuming triples to be independent is a very simplistic way to model the real world***. This is needed to reach your conclusions. Otherwise, given fixed ratio, we can assume complex queries are distributed differently, yielding to another bias in the type numerosity.
> > >
> > > This is basic probability. Assuming two random variables are independent induces a certain joint distribution you are sampling from (the simplest ever). ***If we do not assume independence, we can have that, for instance, if I sample a first missing link, the probability of sampling a second one given (conditioning) the first is much higher than sampling a known link***. This implies that there are relationship in the joint distribution you cannot break by independence, you cannot forget the conditioning part. And it is a more real-world assumption: ***if you do not have information about the first link, why would you have information about a link that depends on it (conditioning)?***
> > >
> > > > Please be aware that the discussion above holds regardless of the sampling algorithm.
> > >
> > > Again, ***this is factually wrong***. We hope we clarified this point for the third time, with an argument from basic probability.
> > >
> > > Lastly, we remark that it is easy to see why the benchmarks are skewed and inflating results, with the current sampling process (which implies triple independence). Do this mental experiment: increase the current ratio of artificially missing edges from 15% to 30%. This scenario still falls under your misunderstood condition that $|E_m| < |E_o|$. Given the current triple independence assumption, suddenly the number of easy query-answer pairs would drop from 98% to 80%, and current average MRR performance for CQA  as reported in papers could drop up to 10-15 points, while the single link prediction one will be remaining stable (up to noise). ***This clearly highlights how the old benchmarks were not designed to measure the spectrum of CQA in mind and why we need more benchmarks***.

---

> > > > ### Comment · Reviewer_z9wR · 2024-12-01
> > > > **Response to "You are assuming triple independence, why is this realistic? (I and II)"**
> > > >
> > > > Dear authors,
> > > >
> > > > Thanks very much for your continuous engagement in the process that we make the question we discussed more precise. If you feel the goalpost was ever moved, the reason might be that you dodged or misinterpreted my questions from time to time. The title of your post "You are assuming triple independence, why is this realistic?" is a perfect example suggesting that you didn't listen to me when I put forward a series of notations.
> > > >
> > > > In this round, I hope I fully understand the misunderstandings. I hope those questions can explain why there is such a radical disagreement between us. And I hope we can face those questions faithfully and let the community and chairs of higher levels judge. I prepare to state the questions here and explain my opinion afterward.
> > > > 1. Is the ratio of missing edges in a KG large or small?
> > > >     - In your opinion: "In a given KG, the number of real missing edges is far greater than the number of seen edges."
> > > > 2. Is the definition of $T_{X, I}, T_{X, II}, T_{X, III}$ and $S_{X, I}, S_{X, II}, S_{X, III}$ independent from sampling?
> > > >     - In your opinion, this is dependent.
> > > >
> > > > ## Q1. Is the ratio of missing edges in a KG large or small?
> > > > This question concerns an axiom-level assumption to the knowledge graphs we handle. You cannot prove or disprove it.
> > > > **I believe the ratio is small** and will be smaller for the knowledge graphs. Here is why:
> > > > 1. Industrial-level knowledge graphs and their embeddings already supported a vast number of applications. If a majority of knowledge cannot be predicted by existing edges and ML models, many applications we saw will not happen.
> > > > 2. In the future, the ratio of missing knowledge will be smaller because
> > > >     1. LLMs that learn from large corpus support question-answering applications quite well with the assistance of existing knowledge graphs or knowledge bases. That means the training corpus of LLMs + existing KG or KB consists of a great amount of knowledge we need.
> > > >     2. Information extraction from the sources above (training corpus of LLMs + existing KG or KB) is easy, given the success of LLM in natural language understanding.
> > > >     3. Based on 1 and 2, I think the ratio of missing edges in KG will be small to satisfy people's real needs.
> > > > 3. Your argument, as I quoted here, is problematic to me.
> > > > > Consider just the proportion of edges in the current FB, NELL or even WikiData, where one can find millions of entities, this implies the possibility to have thousands of billions of possible triples. But only a fraction are observed. So in the real world $|E_m| > |E_o|$.
> > > >
> > > > You assume that there could be pair-wise connections in KG first. **This assumption reflects your belief**. But please see my arguments above, and they imply that if there is a certain amount of knowledge graph edges sufficient **to support people**, we can approach them with the technology by far, so your belief is not correct for **application purpose**. Of course, you can check whether there are pair-wise connections and play the endless game of relation enrichment to connect entities for logical completeness. I am very passive about how active this research direction can be.
> > > >
> > > > ## Q2. Is the definition of $T_{X, I}, T_{X, II}, T_{X, III}$ and $S_{X, I}, S_{X, II}, S_{X, III}$ independent from sampling?
> > > > Let's define it with an algorithmic example.
> > > >
> > > > By saying $T_{2p}$ is a set of **all** reasoning trees for 2p (2-path), I am suggesting that you can obtain $T_{2p}$ by iterating every entity in KG and conducting a DFS with depth of 2. Then you get a set $T_{2p}$. This set contains all possible reasoning trees because it is the largest already, no matter how you sample the data in your script.
> > > >
> > > > For the query type $X$ other than 2p, we always keep $T_{X}$ the largest set, no matter how you sample them.
> > > >
> > > > By saying $S_{X}$ is a set of **all** query-target pairs, I am suggesting that you can populate a query from each reasoning $t\in T_{X}$ and $S_{X}$ contains all of them. So, $S_{X}$ is already the largest set of query-target pairs (because every query-target pair has a reasoning tree), no matter how you sample the benchmark.
> > > >
> > > > Then, each $T$ or $S$ is separated into three splits depending on whether the reasoning tree (or all reasoning trees in a query) is a subgraph of $E_o$ (Type I), $E_m$ (Type III), and otherwise (Type II). Those three splits are non-overlapping by definition.
> > > >
> > > > Please note that for now, we don't need a specific sampler to define those three types. Your comment in the post, "*You are assuming triple independence, why is this realistic? (II)*" exactly reflects that you are unaware of this problem structure.
> > > >
> > > > Then, no matter how you sample the dataset, the dataset you constructed is only contained in $S_{X, III}$ but never $S_{X, II}$. Here is why I think your benchmark does not have sufficient coverage, which totally ignores the performance on $S_{X,II}$.

---

> > > > > ### Comment · Reviewer_z9wR · 2024-12-01
> > > > > **Continued**
> > > > >
> > > > > Whether $S_{X, II}$ is significant or not is determined by both graph topology and the ratio of $E_m$ and $E_o$ (in my belief $|E_m|$ is smaller, but I cannot change your belief so let the future readers decide it.). I want to note that this argument is already free from my previous rough estimation, so please provide further justification if you want to reject it. In addition, if one accepts my belief that $|E_m|$ is smaller and will be even smaller than $|E_o|$, I think it is just natural if two graphs $E_m$ and $E_o$ follow a mildly similar graph property, the counter-example can of course happen for example when $E_m$ is complete and $E_o$ is sparse, but the existence of mild graph property is again, my belief. I believe some empirical counting can be side evidence to justify the size of $S_{X, II}$. I will do that later.

---

> > > > > > ### Author Response · Authors · 2024-12-01
> > > > > > **Ignoring math?**
> > > > > >
> > > > > > Please revise carefully the evidence we provided above with the claims about MRR decrease if you just increase the ratio of artificial missing to 30%. That suffices to say that with the current benchmarks, performance will significantly change for CQA, implying the importance of how many triples one selects, something more than just "believing they can be smaller" as you are doing.

---

> > > > > ### Author Response · Authors · 2024-12-01
> > > > > **Debunking non-factual assumptions**
> > > > >
> > > > > We are trying our best to debunk non factual claims. Most of your comments are ***your beliefs*** as you write yourself. We rebut them in the following, highlighting how there is no concrete evidence to support them. We provided concrete evidence for ours, but even if you think also ours are just beliefs, why yours should be the right ones and should be the motivation to reject a paper?
> > > > >
> > > > > >  This question concerns an axiom-level assumption to the knowledge graphs we handle. You cannot prove or disprove it. I believe the ratio is small [...] Industrial-level knowledge graphs and their embeddings already supported a vast number of applications. If a majority of knowledge cannot be predicted by existing edges and ML models, many applications we saw will not happen. In the future, the ratio of missing knowledge will be smaller because
> > > > >
> > > > > This is wishful thinking. ***You are assuming that FUTURE KGs will be almost complete***, but even if we assume this (very hard to do, see next) this tells us nothing about the fact about the completeness of current KGs, nor about the current benchmarks, FB15k and NELL.
> > > > >
> > > > > It is plainly inconsequential reasoning. It is like saying “we believe future AI will be safe, otherwise many applications we saw did not happen, therefore it is useless to claim some instance of current AI is not safe”.
> > > > >
> > > > > > LLMs that learn from large corpus support question-answering applications quite well with the assistance of existing knowledge graphs or knowledge bases. That means the training corpus of LLMs + existing KG or KB consists of a great amount of knowledge we need.
> > > > >
> > > > > LLMs are notoriously unreliable, and – more crucially – out of the scope of the evaluation of CQA as done not only in our paper, but in twenty and more papers before us.
> > > > >
> > > > > > Based on 1 and 2, I think the ratio of missing edges in KG will be small to satisfy people's real needs.
> > > > >
> > > > > We wish to believe you, but you either need to provide some concrete statistics or you are making some claim about a future that does not impact past benchmarks in any way.
> > > > > > You assume that there could be pair-wise connections in KG first. This assumption reflects your belief.
> > > > >
> > > > > This is not a belief, this is statistics (actually simple math) in action. In ***current real-world KGs we use as benchmarks, only a fraction of links is observed and used to create a dataset***. Consider FB15k-237 it has 14,541 entities and 237 relation types, so potentially 50,111,441,397 triples, but we observe only 310,116 of them. Even if you assume logical constraints (we did not find them in the schema definition),*** there will be billions of missing edges***.
> > > > > You miss an additional crucial point, the ML model does not know in advance which are invalid (do not satisfy logical constraints) or not, it is its job to output a probability score for every possible triple among those 50+ billions. So we have to consider the remaining ones missing.
> > > > >
> > > > > > I am suggesting that you can obtain $T_{2p}$ by iterating every entity in KG and conducting a DFS with depth of 2.
> > > > >
> > > > > There is a profound misconception here: what you are describing here is enumerating all reasoning trees, assuming they are all important. They are not, each one has associated a probability, and this follows an (unknown) joint probability distribution. We need it to i) model uncertainty (otherwise we will not be using ML models) and ii) select a relevant sample of reasoning trees for our dataset. ***There is always a sampling algorithm employed.*** But people not familiar with probabilities are ignoring the fact that the way they sample/construct the training set is dependent on certain (implicit) assumptions.
> > > > >
> > > > > Now, sampling uniformly is impossible because of the large sample space. Consider FB15k-237, as stated above there are 50+ possible triples, and therefore much more 2p queries. As we need a practical way to create a dataset, we need other sampling strategies. Hence people used the triple independence assumption.
> > > > >
> > > > >
> > > > > > Please note that for now, we don't need a specific sampler to define those three types. Your comment in the post, "You are assuming triple independence, why is this realistic? (II)" exactly reflects that you are unaware of this problem structure.
> > > > >
> > > > > ***The current benchmarks are created using a particular sampling strategy, assuming triple independence*** this is not a belief, but concrete evidence (you cannot ignore it!): it is in the code scripts everyone uses. This assumption allows to quickly generate some data, but this data is biased and the distribution of possible queries skewed.
> > > > >
> > > > > > Here is why I think your benchmark does not have sufficient coverage, which totally ignores the performance on $S_{X,II}$.
> > > > >
> > > > > The fact ***we are extending the old benchmarks with new benchmarks only augments the current coverage*** (we are not saying to get rid of the old benchmarks). This profound confusion you have is not a good motivation to reject the paper.

---

> ### Comment · Reviewer_z9wR · 2024-11-30
> **continued responses**
>
> ## Why do I feel this benchmark is not necessary?
>
> Another point you made in your feedback is that, even though only the proposed benchmark is insufficient, it is still meaningful for the readers to see the performance on a particular set of hard samples in $S_{X, III}$ in the so-called stratified analysis.
>
> However, this suggestion is actually not necessary for the following reasons:
> - The hardness of $S_{X, III}$ is caused by the generalization gap of the link prediction. Let's conduct one mental experiment where the link prediction is perfect. Then, the performance drop due to the of the link prediction is gone, which is exactly how the proposed stratified analysis is constructed. The ONLY thing that the stratified analysis can reveal is how other logical elements performed on different parts of the data $S_{X, II}$ or $S_{X, III}$, which was identified as the confounder in your previous response and is not significant in previous practice. This mental experiment suggested that one could almost diminish the gap between $S_{X,II}$ and $S_{X,III}$ queries by directly improving the link prediction.
> - Then, let's consider what the practitioners could do if they see the old averaged scores and new stratified analysis. When they see the old averaged score, what they will try to improve is, as you argued, almost just link prediction. But interestingly, the improved link prediction will also close the gap between $S_{X, III}$ and $S_{X,II}$ performances. Interestingly, the problem you raised can be solved by optimizing the old benchmark, with still sufficient attention on logical elements on $S_{X, II}$. On the other hand, when they see a stratified analysis, they think we might sacrifice some performance on dumb samples in $S_{X,II}$ criticized by you, but win some new points from the hard samples in $S_{X,III}$ encouraged by you. However, this practice is actually encouraging in that the model overfits a tiny portion of datasets, which could be possibly problematic as one loses attention to logical elements that make the queries **really logically complex** on a broad range of data in $S_{X,II}$, as I demonstrated before.

---

> > ### Author Response · Authors · 2024-12-01
> > **Wishful thinking or misunderstanding of logic?**
> >
> > > Let's conduct one mental experiment where the link prediction is perfect. Then, the performance drop due to the of the link prediction is gone
> >
> > This is ***factually wrong***. You can reconstruct the training triple tensor perfectly, but fail terribly at reasoning (this is what ML models are trying to do!), because you do not know how to emulate exactly a formal reasoner in a ML (even hybrid) model.
> >
> > > . But interestingly, the improved link prediction will also close the gap between and performances
> >
> > This is wishful thinking, and perhaps rooted in a logical fallacy: ***improving the accuracy of ML models on hard queries implies improving single link prediction performance***. The opposite is not true.

---

> > > ### Comment · Reviewer_z9wR · 2024-12-01
> > > **Response to "Wishful thinking or misunderstanding of logic?" and "Debunking non-factual assumptions"**
> > >
> > > > This is factually wrong. You can reconstruct the training triple tensor perfectly, but fail terribly at reasoning (this is what ML models are trying to do!), because you do not know how to emulate exactly a formal reasoner in a ML (even hybrid) model.
> > >
> > > This is a mental experiment. As you said, it will never be achieved. The goal of raising this is to highlight the main cause of the hardness of the proposed benchmark, and this point will further be part of my argument that the new benchmark will be unnecessary but potentially confusing. I will explain why next.
> > >
> > > > This is wishful thinking...
> > >
> > > You must have misunderstood my point. Of course, **improving link prediction implies improving CQA** can not be derived from the claim **improving CQA implies improving link prediction**. But my claim is not made by the abovementioned logic.
> > >
> > > Please keep in mind the sufficient condition of perfect CQA performance, which is how the data is sampled.
> > >
> > > *(A) Perfect link prediction* $\land$ *(B) perfect graph traversal* $\land$ *(C) perfect logical calculus* $\to$ *(D) perfect CQA performance*.
> > >
> > > The claim **improving link prediction implies improving CQA** is a relaxed version of the sufficient condition above. The relaxation is that we replace the "perfection" in (A), (B), (C), and (D) by some scores s(A), s(B), s(C), s(D) in [0, 1], 1 means perfect.
> > >
> > > Mathematically speaking, **improving link prediction implies improving CQA** is equivalent s(D) increases with the growth of s(A) up to a noise, with fixed s(B) and s(C).
> > >
> > > First, let's consider QTO, GNN-QE and CQD. For those models, (B) and (C) are almost satisfied by how they store their search stats and conduct the triangular norms. My mental experiments before wanted to suggest that if (A) is satisfied, then (D) is satisfied because (B) and (C) are satisfied too. If you didn't see any flaws here, it means s(D), as a function of s(A), say s(D) = f(s(A)), satisfies f(1) = 1 already. Also, f(0) = 0 is self-evident. So, given a snapshot of model checkpoints where more and more links are successfully predicted, it seems that f(x) is an increasing function up to a noise.
> > >
> > > For general neural models, (B) and (C) are only partially satisfied but fairly good, as suggested by your table A.3. The link predictors should work well for both prediction and collaboration with other components implementing functions in (B) and (C). The general trend, however, is still there and might be revealed by a regression plot. Is it also wishful thinking? I am curious about your opinion.
> > >
> > > -----
> > >
> > > # Response to "Debunking non-factual assumptions"
> > >
> > > We still have different beliefs that distort our sense of the importance of various parts of data, such as you want to stress hard benchmarks, and I think distorting $S_{X,II}$ is also necessary. I believe this is why we need other reviewers. I would like to leave the right and wrong to the others to decide and promise to weaken this impact on the final score.
> > >
> > > I need to emphasize that this very difference in belief does not sufficiently confuse me. My position is that this new setting is somehow invalid because of the combination of two arguments: the first one is the attention paid to different parts of data. I think we already agreed; the second one is that it is confusing when working with old benchmarks, especially if a better link prediction solves it; as I described earlier, what is the value added by evaluating the new benchmark that conveys the old message?

---

> > > > ### Author Response · Authors · 2024-12-01
> > > > **Non sequitur?**
> > > >
> > > > > This is a mental experiment. As you said, it will never be achieved.
> > > >
> > > > We fail to see the value of this mental experiment that will never be achieved and follows some "interesting" non-classical logic. We comment more below.
> > > >
> > > > > You must have misunderstood my point. Of course, improving link prediction implies improving CQA can not be derived from the claim improving CQA implies improving link prediction. But my claim is not made by the abovementioned logic. [...] The claim improving link prediction implies improving CQA is a relaxed version of the sufficient condition above
> > > >
> > > > We fail to understand what this made-up relaxation of a non-sufficient condition can imply rigorously. Please, ***let's stick to rigorous logical reasoning***, or bring concrete evidence when criticizing precise claims.
> > > >
> > > > > First, let's consider QTO, GNN-QE and CQD. For those models, (B) and (C) are almost satisfied by how they store their search stats and conduct the triangular norms
> > > >
> > > > This is again factually wrong, ***there is no concrete evidence that B and C are "almost satisfied***, but there is plenty of evidence of the opposite. In fact, if it were true, then QTO, GNN-QE and CQD would perform much better for simple queries such as 2p on the new benchmarks. Here you can have a reason why the new benchmarks are important: you can exactly disentangle A, from B and C.
> > > >
> > > > >  If you didn't see any flaws here, it means s(D), as a function of s(A), say s(D) = f(s(A)), satisfies f(1) = 1 already. Also, f(0) = 0 is self-evident. So, given a snapshot of model checkpoints where more and more links are successfully predicted, it seems that f(x) is an increasing function up to a noise.
> > > >
> > > > We see several flaws in this reasoning. You are projecting what you would like to see and achieve in a scenario that has no concrete basis to exist. The experimental evidence -- ***please re-run our experiments*** -- (and see also our answer to z2gA) says the opposite. Here you are claiming that the "growth" of any of these property is also the same for all of them. We wish this could be the case -- *hence all CQA would be easily solvable by link prediction and no more papers on complex reasoning are necessary* -- but rigorous empirical evidence claims the opposite.
> > > >
> > > > > For general neural models, (B) and (C) are only partially satisfied but fairly good, as suggested by your table A.3. The link predictors should work well for both prediction and collaboration with other components implementing functions in (B) and (C). The general trend, however, is still there and might be revealed by a regression plot. Is it also wishful thinking? I am curious about your opinion.
> > > >
> > > > Yes it is wishful thinking unfortunately, as Table A.3 highlights the opposite, the ***MRR suddenly drops already for the `2p` column***. Where do you see in it that `(B) and (C) are only partially satisfied but fairly good`?
> > > >
> > > > > We still have different beliefs that distort our sense of the importance of various parts of data
> > > >
> > > > We provided solid math evidence on the number of missing links being much larger on real-world KGs. Could you please comment on that? Do you really believe that among the 50+ billion possible triples of FB15k-237 there are less "truly" missing links than 200k observed ones?. Even if you (arbitrarily) decide that 49 billion triples are meaningless somehow, there will be more missing links in the remaining 1+ billion.

---

> ### Comment · Reviewer_z9wR · 2024-12-02
> **Response in to "Non sequitur?"**
>
> Dear authors,
>
> I am glad to see our discussion converge to a certain topic, indicating the debate will end soon. According to your post, I think now the frontier is about (1) Why I am saying the hardness of your hard class is caused by link prediction and can be fixed by the link predictor and (2) the "belief" about whether the ratio of missing edges in KGs is large or small.
>
> ## About (1)
>
> > We fail to see the value of this mental experiment that will never be achieved and follows some "interesting" non-classical logic.
>
> In fact, you have just simulated this mental experiment when you sample your new benchmark. You see all the missing edges in your sampling algorithm. You use the derived results as the gold answer.
>
> > This is again factually wrong, there is no concrete evidence that B and C are "almost satisfied
>
> I don't need empirical evidence for that.
> - For (B), please check the computation of QTO, and you should realize that QTO calculated graph traversal by multiplication of the adjacency matrix produced by the link predictor. (B) holds if you admit that the matrix multiplication can simulate graph traversal.
> - For (C), please check the definitions of triangle norms. (C) holds if you admit that they collapse into the standard logic calculus when the input truth value is boolean.
>
> If you don't like my wording, what would you call it if a computation process were implemented as its definition?
>
> > In fact, if it were true, then QTO, GNN-QE and CQD would perform much better for simple queries such as 2p on the new benchmarks.
>
> On my side, I think we can blame the link predictor because the algorithm implemented by QTO/FIT follows the standard evaluation of existential queries if the adjacency matrix used by them is perfect (by link predictor is perfect, but this does not happen usually). Please check Chapter 4 in the literature [8].
>
> Also, please explain your statement if you want to use it.
>
> [8] Suciu, D., Olteanu, D., Ré, C., & Koch, C. (2022). Probabilistic databases. Springer Nature.
>
> > Here you can have a reason why the new benchmarks are important: you can exactly disentangle A, from B and C.
>
> I don't understand this sentence. Please elaborate.
>
> > We see several flaws in this reasoning. You are projecting what you would like to see and achieve in a scenario that has no concrete basis to exist.
>
> As I said before, this scenario is exactly how you produced your benchmark, which suggested one shortcut to exceed in your benchmark. It is okay if you are happier to call it unrealistic. But please also look at some related arguments.
>
> > The experimental evidence -- please re-run our experiments -- (and see also our answer to z2gA) says the opposite.
>
> Please see my above and explain what is the reason if I can not produce the correct answer only because I had a bad adjacency matrix produced by the imprecise link predictor.
>
> ---
> ## About (2)
>
> > We provided solid math evidence for the number of missing links being much larger on real-world KGs.
>
> **I have no intention of changing your personal beliefs because this is your free will, and I left it to the community’s judgment. I stated that the impact of the disagreement in belief would be weakened in the final score**. But if you look into FB15k-237 for solid math evidence for your belief, here is my response because this is the real case I can analyze.
>
> Let me recall your math evidence by quoting your earlier responses as follows:
>
> > Consider FB15k-237 it has 14,541 entities and 237 relation types, so potentially 50,111,441,397 triples, but we observe only 310,116 of them.
>
> And also the following quote:
>
> > Even if you (arbitrarily) decide that 49 billion triples are meaningless somehow, there will be more missing links in the remaining 1+ billion.
>
> Please add to it if I missed anything about your math.
>
> You must want to distinguish two concepts: (1) the total number of possible triples. (2) the missing triples we care about, that is, the triples can be justified as true. Recall that knowledge is justified as true.
>
> **Please look at the actual data, in particular the relations, in FB15k-237 before you tried to persuade yourself with those evidences**
>
> You can look at your data or the public source I mentioned below. https://github.com/liuxiyang641/RAGAT/blob/main/data/FB15k-237/rel_cat.ipynb
>
> Please go through every relation and ask some questions like the following examples.
> - How many triples can be found for relation "/people/person/place_of_birth"? How many places of birth can one person have? Is it 14,541?
> - How many triples can be found for relation "/award/award_winning_work/awards_won"? How many movie awards have been made in human history? Is it more than 1% of 14,541*14,541?
>
> You can then see that your estimation is inflated by (1) counting triples with impossible relation types. (Can entity pair (a movie, an award) be connected by "/people/person/place_of_birth"?) (2) ignoring the fact that many relations are just sparse.

---

> > ### Author Response · Authors · 2024-12-03
> > **Final debunking?**
> >
> > > On my side, I think we can blame the link predictor because the algorithm implemented by QTO/FIT follows the standard evaluation of existential queries if the adjacency matrix used by them is perfect (by link predictor is perfect, but this does not happen usually). Please check Chapter 4 in the literature [8].
> >
> >
> > You mentioned `QTO, GNN-QE and CQD` first and now you are providing some ex-post explanation only for QTO. ***This is goalpost shifting*** and it happened several times, starting from the first review that only criticizes the paper about missing baselines and query types. ***Empirical evidence is rejecting your claim that improving link prediction is sufficient to improve CQA***, as follows.
> >
> > First, GNN-QE and CQD do not even try to approximate Dan Suciu’s algorithm, as QTO does, and yet they can perform better than QTO in many scenarios (see our Tables on new and old benchmarks). This is even more striking when we re-run QTO using the same link predictor we use for CQD. ***You cannot blame the link predictor alone***, as CQD with the same link predictor and a much simpler algorithm fares better than QTO. Your approximation of Dan’s algorithm in QTO, if it were true reasoning, would score better than the cruder algorithm in CQD. This is not the case, and we provided reproducible code.
> >
> > Furthermore, you are confusing formal reasoning with probabilistic reasoning, even if you were to implement exactly Dan’s algorithm with perfect numerical precision, you would not be guaranteed to predict the correct answers, but just the correct distribution. When you move to the mode of the distribution your performance can drop. Another hint that the current benchmarks can distort the sense of performance gains we have.
> >
> > > Please see my above and explain what is the reason if I can not produce the correct answer only because I had a bad adjacency matrix produced by the imprecise link predictor.
> >
> > You are confusing an adjacency matrix with a probability tensor (that is definitely non-sparse, unless you enforce constraints as in [A]). And confusing logical reasoning with probabilistic reasoning. Current score-based models are not giving you (calibrated) probabilities [A].
> >
> > [A] Loconte, Lorenzo, et al. "How to turn your knowledge graph embeddings into generative models." Advances in Neural Information Processing Systems 36 (2024).
> >
> > > I have no intention of changing your personal beliefs because this is your free will, and I left it to the community’s judgment.
> >
> > It is not a matter of changing beliefs, it is supporting beliefs with solid evidence. In case you do not have evidence for yours, you should at least admit “I do not know” instead of claiming the others’ are wrong. If they are equivalently probable, you have no reason to reject.
> >
> > > Please add to it if I missed anything about your math.
> > Yes, please re-read again what we wrote above. You have to take logical constraints into account, as we wrote above and therefore the number of valid triples drops from 50+ billions. However, even if the number were to drop by 99.99%, you would still have 5 million triples. Much larger than the current observed number of 200k triples.
> >
> > As a last remark, we are quite baffled that simple math and logical arguments can be taken lightly. ***The point of a review process is not to impose personal beliefs about future KGs or speculative opinions about what models are doing**, it is to exchange solid evidence about what we observe and what we can quantify properly.
> > This is one of the aims of our paper: properly and rigorously measuring what we are reporting in tables, disentangling the hype from science.

---

> ### Comment · Reviewer_z9wR · 2024-12-03
> **How can you debunk with logical fallacy?**
>
> > You mentioned QTO, GNN-QE and CQD first and now you are providing some ex-post explanation only for QTO. This is goalpost shifting.
>
> I think QTO is a valid example that made my arguments clear due to my limited time. But it seems that I failed :-). But thanks for your response and for being OKAY with my justification with QTO. Then, let me express my point why CQD and GNN-QE are also satisfied.
>
> 1. For (B), the link predictor part. I say that QTO uses the adjacency matrix multiplication, and the link predictor is perfect if the **adjacency matrix produced by scores** is perfect, and added to that, **set projection modeled by adjacency matrix multiplication**. Now, you should see that the link predictor plays a crucial role with the combined (i) and (ii), which means the **perfect set projection**.
>   - Now you should look at CQD, I suppose you used CQD-beam, if you can just make your beam size larger, the argument is the same as QTO.
>   - If you look at GNN-QE, you should be aware that the the role of the backbone NBFNet plays the role exactly as **set projection**, which is made by a perfect link predictor.
>
> 2. For (C), both use triangle norm.
>
> > Furthermore, you are confusing formal reasoning with probabilistic reasoning. [...] You are confusing an adjacency matrix with a probability tensor (that is definitely non-sparse, unless you enforce constraints as in [A])
>
> Why do the link predictor produce a dense adjacency matrix? This very accurate phrase is named by its functionality of predicting links. A formal reasoner, if it predicts links, can also be named a link predictor. I am not talking about KG embedding; please read carefully. What do you mean by a paper about KG embedding? Don't **distort my words**.
>
> When a link predictor is perfect, the adjacency can be binary. I hope you can see that those two concepts (formal reasoning with probabilistic reasoning) are the same under this very circumstances, and this very discussion is also linked to the mental experiment, and also how you construct your hard dataset.
>
> > it is supporting beliefs with solid evidence.
>
> I am very cautious with your example. Your reasoning is flawed; I can only see what you are doing is like (1) the number of total candidates is large, but that is okay. (2) Let's assume there is a constant ratio of the missing links; if 1/49 does not seem convincing enough, let's try 1/10000.
>
> Do you think (1) and (2) is valid?
>
> But wait, why is the ratio a constant? Have you ever thought about that critically?
>
> ----
>
> This is a friendly reminder. In your revised manuscript, the equation (1) about the definition of MRR is wrong; it seems to be a mean rank. :-)

---

> > ### Author Response · Authors · 2024-12-03
> >
> > >  Now, you should see that the link predictor plays a crucial role with the combined (i) and (ii), which means the perfect set projection.
> >
> > you are assuming you have perfect B and C, which is not true. You started by saying you wrote a mistake, now you are saying again that it is a perfect reasoning. It is not. And as we said above, ***the proof is in the pudding: the empirical scores are supporting our claim***. In fact...
> >
> > > Now you should look at CQD, I suppose you used CQD-beam, if you can just make your beam size larger, the argument is the same as QTO.
> >
> > ... the simple CQD-beam is better than QTO in many scenarios, see our tables and this happens with having the same link predictor. Can you comment on that? Why is QTO worse? No need to increase the beam size for CQD. Hence, the B and C are not perfect in QTO.
> >
> > > If you look at GNN-QE, you should be aware that the the role of the backbone NBFNet plays the role exactly as set projection, which is made by a perfect link predictor.
> >
> > also ***the implementation in GNN-QE is not realizing perfect logical (or probabilistic) reasoning***. The backbone can not implement Dan's algorithm. No need to debate this. In any case, you are building now a wishful argument to defend your stance that link prediction is sufficient and that we do not need additional benchmarks. ***But by the same line of reasoning, you would not need the old CQA benchmarks, and just link prediction benchmarks.***
> >
> > > When a link predictor is perfect, the adjacency can be binary.
> >
> > This is wishful thinking, mistaking for "perfection" something that will never happen in ML, and something we do not want in probabilistic ML. There is no easy way to learn -- in our current differentiable pipelines -- a perfect 0 and 1, unless we manually crop/clip the probability values ex-post.
> >
> > > Do you think (1) and (2) is valid?
> >
> > No, as you are setting the ratio based on the observed triples we have and on the test split ratio people arbitrarily set for the old benchmarks, which were defined for link prediction, and not CQA. ***We state this again, believing that link prediction alone is sufficient for CQA is misleading and factually not true.***
> >
> > > But wait, why is the ratio a constant? Have you ever thought about that critically?
> >
> > The ratio is a constant: a quotient between two scalars. An Unknown constant. Your flawed assumption is that this is known and based on the artificial split.
> >
> > > This is a friendly reminder. In your revised manuscript, the equation (1) about the definition of MRR is wrong; it seems to be a mean rank. :-)
> >
> > Thanks for finally spotting it (why a reminder?), we can easily fix it.

---

### Author Response · Authors · 2024-11-20
**Global answer to all reviewers**

Dear reviewers,

Thanks for the insightful feedback and questions. Before addressing your questions individually, we would like to remark the following points.

**Scope and motivation.** We remark that ***our main contribution is not to propose a new method for CQA, but to reveal that the reported performance of standard benchmarks are inflated due to the presence of training links***. Table 1 helps understand our claim: very few queries are full-inference, and the aggregate performance people report in papers (even QTO, see below), is mostly due to the ability to memorize links and solve (1-hop) link prediction tasks. While we analyze only positive queries, our analysis can be easily transferred to negative queries and to benchmarks beyond FB15k-237 and NELL995, as the presence of training links depends on *how* a benchmark is created, rather than on the specific query types it contains.

**Negation queries.** In fact, analyzing the queries involving negation in FB15k-237 and NELL995, reveals training links in the non-negative reasoning tree of the query-answer pairs (as expected). From our counts, 95.4% of 3in query-answer pairs and 98.4% of pin in FB15k-237 have existing links present in the non-negative part of their reasoning tree.
Furthermore, also negated links appear in the training, thus potentially leaking information (how this propagates to performance is less clear than the positive-part, as each method treats negation differently). We report these values in Table A.1 in Appendix A.1.
We are running the full analysis for all remaining query types and will provide the remaining results in the revised version of the paper and in this discussion.
The bottom line is: also ***negated queries are affected by the leak*** and potentially all queries that are generated in the same way.


**QTO and hybrid solvers.** We were not aware of the existence of other hybrid solvers in the literature. We thank the reviewers for pointing us to QTO and FIT. We amended the line “To the best of our knowledge, this is the first time such an hybrid solver is proposed in the CQA literature.” and added more lines to contextualize QTO on page 8. We remark that CQD-Hybrid is not the main contribution of the paper, see above.

We introduce CQD-Hybrid to test our hypothesis that performance on the old benchmarks are inflated. We support our claim by showing that even the remarkable performance reported by QTO in its original paper is also an artifact of the fact that ~98% of all test queries require predicting a single link (see Table 1) as it happens for all baselines.
In fact, if we run our stratified analysis for FB15k-237 with QTO to as we do in Table 2, it follows the same pattern of the other methods:


| Query type | all | 1p    | 2p   | 3p   | 2i   | 3i   | 1p2i | 2i1p |
|------------|---------|-------|------|------|------|------|------|------|
| 1p         | 46.7    | 46.7  | -    | -    | -    | -    | -    | -    |
| 2p         | 16.6    | 16.7  | 4.0  | -    | -    | -    | -    | -    |
| 3p         | 15.6    | 15.8  | 4.5  | 5.0  | -    | -    | -    | -    |
| 2i         | 39.7    | 40.8  | -    | -    | 5.7  | -    | -    | -    |
| 3i         | 54.6    | 56.4  | -    | -    | 15.4 | 5.4  | -    | -    |
| 1p2i       | 33.8    | 35.9  | 15.8 | -    | 6.2  | -    | 7.3  | -    |
| 2i1p       | 24.7    | 25.3  | 10.3 | -    | 8.6  | -    | -    | 8.1  |



i.e for 2p queries: the aggregate statistic reported in the paper (‘all’) has mrr=16.6, and 2p queries that are reducible to 1p have mrr=16.7. But 2p queries that are not reducible to any query type have mrr=4.0, much lower. We report the complete values for FB15k-237 and NELL995 in the updated table A.3 and A.4.

This highlights that the issue is to be found not in “old’’ baselines we use, but in the classical benchmark and affects all baselines.

**QTO on new benchmarks.** We also evaluate QTO on our new benchmarks, and we find that performance indeed degrades as for the other baselines. We are still running the last experiments and we will report full data in Table 5. Here is an extract of the table for FB15k-237+H:

| Model        | 1p    | 2p    | 3p    | 2i    | 3i    | 1p2i  | 2i1p  | 2u    | 2u1p  |
|--------------|-------|-------|-------|-------|-------|-------|-------|-------|-------|
| GNN-QE   |42.8 | **6.5** | **4.2** | 10.3  | 10.3  | 4.6 | 7.8   | 36.9  | 7.2 |
| ULTRAQ   | 40.6  | 5.5   | 3.7   | 8.2   | 7.9   | 4.1   | 7.9  | 33.8  | 5.1   |
| CQD      | **46.7** |6.3 | 2.7   | **18.4** | **16.2** | **5.6** | **9.4** | 34.9 | 7.3   |
| ConE     | 41.8  | 5.7   | 3.9 | 16.8 | 13.9 | 4.0   | 6.6   | 25.9  | 5.3   |
| QTO      | **46.7** | 5.9   | 3.5   | 13.5  | 11.8  | 4.7   | 8.8   | **37.3** | **7.4** |

---

> ### Author Response · Authors · 2024-11-26
> **Follow up - Global answer to all reviewers**
>
> Dear reviewers,
>
> We uploaded a new version of the paper, where we completed all the experiments:
>
> **Negation queries**: We run the full analysis for all negation queries, ***confirming that even by just looking at the positive reasoning tree, the vast majority of query-answer pairs contain existing links*** thus pointing at the same issues we already analyzed for positive-only queries in the paper.  We report these values in Table A.1 in Appendix A.1.
>
> **QTO on old benchmarks**: The additional experiments on the old benchmarks with QTO, table A.3, A.4, ***confirm that QTO aggregated performance is due to 1p queries and it drops for the portion of full-inference query-answer pairs, as any other solver***. Moreover, even when QTO is the SoTA on a certain query type, most of the time it is not so on the portion of full-inference query-answer pairs only, showing that improving performance on the partial-inference query-answer pairs does not necessarily result in improvements over the full-inference ones.
>
> **QTO on new benchmarks**: We completed the experiments for QTO on the new benchmarks, see Table 5. The experiments show that QTO have similar performance with CQD, a much simpler and older baseline, enforcing our claim for which the perception of progress in the field has been inflated due to the presence of a massive amount of partial-inference query-answer pairs in the old benchmarks.

---

### Comment · Area_Chair_URtb · 2024-11-27

Dear reviewers,

Thank you for your efforts reviewing this paper. Can you please check the authors' latest response and see if your concerns have been addressed? Please acknowledge you have read their responses. Thank you!

---

### Meta-Review · Area_Chair_URtb · 2024-12-19

**Metareview:**

This paper shows that current benchmarks for CQA are not really complex, and that in these benchmarks most queries (up to 98% for some query types) can be reduced to simpler problems, e.g., link prediction, where only one link needs to be predicted. The performance of state-of-the-art CQA models drops significantly when such models are evaluated on queries that cannot be reduced to easier types. Thus, the results reported on existing CQA benchmarks might distort researchers’ perception of progress in this field. This paper then proposes a set of more challenging benchmarks, composed of queries that require models to reason over multiple hops and better reflect the construction of real-world KGs, and show that there is a lot of room to further improve current CQA methods on the new benchmarks.

Strengths:

Reviewers generally agree that this paper provides an in-depth study and analysis of existing benchmarks and offers interesting observations.

Weaknesses:

Reviewers actively engaged in discussions with the authors, and many comments raised in the original reviews were addressed through the rebuttal period. Here is a summary of the unresolved (or partially resolved) issues after rebuttal:

1. The distribution of the proposed benchmark by Reviewer z9wR (relatedly, the discussion around query types raised by Reviewer fPSh), and how this impacts the proposed benchmark’s usefulness (or, why it is necessary) still remain unclear.

While I really appreciate the authors’ efforts in actively responding to reviewers’ comments, there are still issues that remain unresolved after the discussion period. In addition, some issues that did get resolved in this process will need to be clarified in the revised version. I believe the paper could benefit from another round of revision which integrates the discussions during the rebuttal period, particularly addressing the necessity of the proposed benchmark.

**Additional Comments On Reviewer Discussion:**

(Partially) Addressed weaknesses:

Most reviews mentioned the lack of baselines such as QTO and FIT originally, which the authors added during the rebuttal period. However, the newly added results of QTO seem problematic to Reviewer z2gA, who even tried to reproduce the results but encountered some issues. I did not consider this to be a major issue in making my decision, but do encourage the authors to consider doing more comprehensive experiments using these baselines in the revised version.

---

### Decision · Program_Chairs · 2025-01-22

Reject